# BAF complex-mediated chromatin relaxation is required for establishment of X chromosome inactivation

Andrew Keniry [1,2 ✉], Natasha Jansz [1,2], Linden J. Gearing [1,2], Iromi Wanigasuriya[1,2], Joseph Chen[3,4,5], Christian M. Nefzger [3,6], Peter F. Hickey [1,2], Quentin Gouil [1,2], Joy Liu[1,2], Kelsey A. Breslin [1], Megan Iminitoff[1,2], Tamara Beck[1], Andres Tapia del Fierro [1,2], Lachlan Whitehead [1,2], Andrew Jarratt[1,2], Sarah A. Kinkel[1,2], Phillippa C. Taberlay [7], Tracy Willson[1,2], Miha Pakusch[1], Matthew E. Ritchie [1,2], Douglas J. Hilton[1,2], Jose M. Polo [3,4,5] & Marnie E. Blewitt [1,2 ✉]

The process of epigenetic silencing, while fundamentally important, is not yet completely understood. Here we report a replenishable female mouse embryonic stem cell (mESC) system, Xmas, that allows rapid assessment of X chromosome inactivation (XCI), the epigenetic silencing mechanism of one of the two X chromosomes that enables dosage compensation in female mammals. Through a targeted genetic screen in differentiating Xmas mESCs, we reveal that the BAF complex is required to create nucleosome-depleted regions at promoters on the inactive X chromosome during the earliest stages of establishment of XCI. Without this action gene silencing fails. Xmas mESCs provide a tractable model for screen-based approaches that enable the discovery of unknown facets of the female-specific process of XCI and epigenetic silencing more broadly.

[1] The Walter and Eliza Hall Institute of Medical Research, Parkville, VIC, Australia. [2] The Department of Medical Biology, University of Melbourne, Parkville, VIC, Australia. [3] Department of Anatomy and Developmental Biology, Monash University, Clayton, VIC, Australia. [4] Development and Stem Cells Program, Monash Biomedicine Discovery Institute, Clayton, VIC, Australia. [5] Australian Regenerative Medicine Institute, Monash University, Clayton, Victoria, Australia. [6] The University of Queensland, Institute for Molecular Bioscience, St Lucia, QLD, Australia. [7] School of Medicine, College of Health and Medicine, University of Tasmania, Hobart, TAS, Australia. ✉email: keniry@wehi.edu.au; blewitt@wehi.edu.au

pigenetic gene silencing facilitates cell-type-specific transcriptional signatures and is therefore fundamental to shaping cell identity in both development and disease. The silencing process is necessarily complex, multilayered, and not fully understood despite significant research efforts. X chromosome inactivation (XCI) is the mammalian compensation mechanism that ensures equal gene dosage between XX females and XY males, resulting in near-complete silencing of one female X chromosome[1–4]. XCI is therefore a powerful system in which to study epigenetic silencing across hundreds of loci in parallel.

In vitro, female mouse embryonic stem cells (mESCs), like the blastocyst cells from which they derive, have activity from both X chromosomes; a feature exclusive to these and primordial germ cells[5–7]. Upon differentiation mESCs undergo XCI creating an active (Xa) and an inactive (Xi) X chromosome in an apparently random process. XCI occurs stepwise after initiation by upregulation of the long non-coding RNA *Xist*. This heralds the establishment phase of XCI, where *Xist* spreads *in cis* to coat the future Xi[8,9] and recruits factors[10–12] that establish silencing through loss of activating[10,13–15] and gain of repressive histone marks[14,16–26] and adoption of a unique bipartite chromosome conformation[27–30], mediated in part by Smchd1[31–33]. Silencing is then maintained by DNA methylation at promoter elements[13,34], complemented at a subset of genes by H3K9me3[23,35]. This rich understanding is the result of three decades of exceptional research that has contributed significantly to our understanding of epigenetic silencing. However, we still do not completely understand the process and as such, XCI remains a fertile system to identify unknown facets of silencing.

Differentiating female mESCs present as an enticing system in which to study XCI; however, complications with in vitro maintenance of these cells have severely limited their use. In culture female mESCs are epigenetically unstable, displaying global hypomethylation compared with males[36–42] and karyotypic instability, with XO cells rapidly dominating cultures[36,40,41]. Based on a desire to study XCI in normal differentiating female mESCs, we created X-linked fluorescent reporter alleles (Xmas), allowing efficient monitoring of karyotype and XCI status in live cells and with minimal manipulation of these sensitive cells. Xmas mESCs allowed us to perform a genetic screen, which although targeted and small in scale, is an initial screen for regulators of the establishment of XCI in its native state; revealing a role for the nucleosome remodellers Smarcc1 and Smarca4. Smarcc1 creates an accessible future Xi that allows XCI to proceed. Therefore, chromatin relaxation may be an initial step in epigenetic gene silencing, demonstrating the utility of the Xmas system for screening in normal female mESCs and subsequent discovery of unknown aspects of XCI.

## Results

### Creation of Xmas reporters that reflect normal random XCI.
To create a tractable and replenishable female mESC system, we knocked either a GFP or mCherry reporter cassette into the 3′ UTR of the X-linked house-keeping gene *Hypoxanthine guanine phosphoribosyltransferase* (*Hprt*, $X^{Hprt\text{-}GFP}$ and $X^{Hprt\text{-}mCherry}$, Fig. 1a). We chose fluorescent reporters to enable the use of fluorescence-activated cell sorting (FACS) to purify live cells with different X inactivation states, thus permitting multiple options for screen-based approaches. We chose to drive reporter expression from the endogenous *Hprt* promoter to give the most natural representation of X-linked expression and silencing possible. Dual reporter systems for XCI studies have been created previously; however, these did not use an endogenous promoter[35,43,44] or did not create lines of mice[40,45], features which would enable the study of XCI in female mESCs in the most native state possible. Our reporter alleles were initially

created in Bruce4 XY mESCs, then to ensure we could continually derive XX mESCs, we created two homozygous/hemizygous mouse strains from the reporter alleles which when crossed produce female offspring with GFP and mCherry marking different X chromosomes ($X^{Hprt\text{-}GFP}$ $X^{Hprt\text{-}mCherry}$, Fig. 1b, Extended Data Fig. 1a). We call this the Xmas (X-linked markers active silent) system. We inserted an internal ribosome entry site (*IRES*) between the *Hprt* stop codon and the reporters, which diminished fluorophore intensity, yet is necessary to ensure appropriate *Hprt* function. The neomycin cassette was deleted to allow detectable fluorophore expression. Roughly equal male and female pups were born of each genotype (Supplementary Fig. 1b–e). Flow cytometry of white blood cells from Xmas animals as well as hematopoietic stem and progenitor cells (LSK) and primary mouse embryonic fibroblasts (MEFs) from Xmas embryos showed the reporter alleles accurately detect random XCI with approximately half the cells positive for each fluorescent protein (Fig. 1c–f).

### Xmas induced pluripotent stem cells (iPSC) and mESCs show two active X chromosomes.
Next, we wished to assess whether pluripotent Xmas cells would display the expected expression of both the Cherry and GFP reporters. Since reactivation of the Xi is a feature of late-stage cellular reprogramming[46], we first tested whether our reporter alleles performed as expected during iPSC induction. Indeed, ~80% of post-XCI $X^{Hprt\text{-}GFP}$ $X^{Hprt\text{-}mCherry}$ MEFs transduced with a doxycycline-inducible reprogramming cassette (STEMCCA, Fig. 2a)[47] detectably reactivated their Xi by the final day of the assay (Fig. 2b). These data show that pluripotent Xmas cells display the expected biallelic expression from the X chromosome and suggest that this system may be a useful tool for studying reprogramming.

We next assessed the suitability of our mouse lines for the production of Xmas mESCs ($X^{Hprt\text{-}GFP}$ $X^{Hprt\text{-}mCherry}$, Fig. 2c, Supplementary Fig. 2a, b). Female blastocysts displayed reporter expression exclusively from the maternal X chromosome in extraembryonic cells, as expected due to imprinted XCI in trophecotoderm[48]. By contrast, both X chromosomes were active in the inner cell mass, indicating the expected reactivation of the silent paternal X chromosome in embryonic cells (Fig. 2d). Following derivation in serum-free, feeder-free conditions with inhibitors of MEK and GSK3 (2i), expression of both reporters was detectable in Xmas mESCs by microscopy and flow cytometry (Fig. 2e, f). However, the abundance of single positive Xmas mESCs progressively increased in culture, likely reflecting the increasing abundance of XO cells. We tested whether the reporters accurately reflected karyotype in mESCs by FACS followed by PCR of genomic DNA and found cells single positive for the reporters were also single positive for the corresponding allele (Supplementary Fig. 2c). Thus, the fluorescent reporters in the Xmas system detect XX and XO mESC populations. This is a very useful feature of the Xmas system as it enables rapid and regular monitoring of the karyotype of female mESCs and offers the opportunity to purify XX cells by FACS to ensure suitability for XCI experiments and minimising confounding results that can occur due to the presence of XO cells.

To assess the similarity of Xmas mESCs to published mESC lines, and therefore their suitability for functional XCI and pluripotency studies, we compared Xmas mESC transcriptomes to previously published data sets of both naive and primed mESCs[49,50]. We found similar expression of pluripotency genes in all three groups, but lower early differentiation-associated genes compared with primed mESCs (Fig. 3a). Xmas mESCs most closely resemble naive mESCs maintained in 2i media (Fig. 3b)[51], as expected given that Xmas mESCs were also derived and

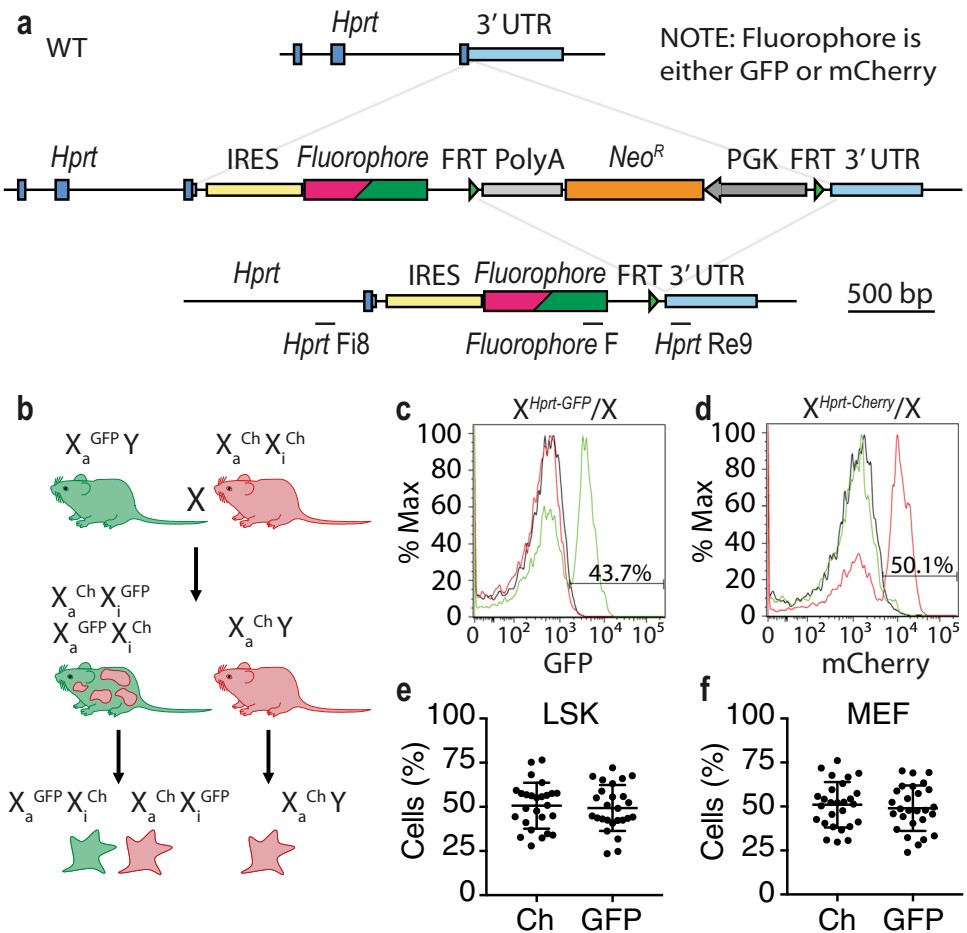

**Fig. 1 Creation of Xmas reporter alleles and strains of mice. a** Schematic of the Xmas reporter alleles indicating the knockin of the reporter into the 3′ untranslated region (UTR) of *Hprt*. Note that fluorophore indicates either GFP or mCherry, with each element cloned similarly. The Flipase recognition target (FRT) and internal ribosome entry sites (IRES) are indicated, as are the genotyping oligonucleotides. **b** Schematic of homozygous/hemizygous reporter allele mouse strains and their XCI status. **c**, **d** Flow cytometry data showing detection of the GFP (**c**) and mCherry (**d**) fluorescent reporters from white blood cells of X^Hprt-GFP/X (green) and X^Hprt-mCherry/X (red) female mice, compared to XY (black) male mice. **e**, **f** Flow cytometry data showing the percentage of each fluorescent reporter allele from ex vivo haematopoietic stem and progenitor cells (LSK) *n* = 26 independent replicates (**e**) and primary MEFs *n* = 26 independent replicates. **f** Error bars show mean ±standard deviation. Source data are provided as a source data file.

maintained in 2i media. Xmas mESCs also retain pluripotency, as they form teratomas that differentiate into all three germ layers (Fig. 3c). To assess in vitro differentiation potential of Xmas mESCs, we induced differentiation by weaning from 2i into serum-containing LIF-free media over 3 days to induce a non-directed differentiation. We performed RNA-seq daily for 9 days during the differentiation. As expected, Xmas mESCs begin most transcriptionally aligned to naive mESCs, transitioning through a primed state, before finally more closely resembling MEFs than neural stem cells (Fig. 3d and Supplementary Data 1), likely reflective of the non-directed differentiation. These data show that Xmas mESCs display similar properties to other naive mESCs in vitro.

**Xmas mESCs detect impaired XCI**. We next tested whether our Xmas reporter alleles could detect random XCI in differentiating mESCs, using the same differentiation protocol as above (Fig. 2c). At the induction of differentiation, most cells were double positive for the fluorescent markers, indicating an XaXa XCI state, before the rapid loss of double positivity from days 5 to 7 with all lines becoming XaXi, as expected following random XCI (Fig. 4a). To test whether this reflected normal XCI timing we derived F1

female mESCs from FVB/NJ (FVB) dams crossed to CAST/EiJ (CAST) sires. Allele-specific analyses are enabled by single nucleotide polymorphisms between each allele and natural skewing of XCI, with the FVB allele approximately three times more likely to become the Xi upon random XCI. This avoids the need to genetically skew random XCI by *Xist* deletion, allowing the most normal process to occur and minimum manipulation of the cells, producing more consistent results in our hands. Allele-specific RNA-seq during F1 mESC differentiation showed *Hprt* follows similar XCI kinetics to other X-linked genes (Fig. 4b) and similar kinetics to our Xmas *Hprt* reporters (Fig. 4a). Notably, Xmas offers the advantage of a single-cell analysis of XCI. Therefore, Xmas mESCs allowed us to consider functional XCI studies.

We tested if XCI could be perturbed in Xmas mESCs by knockdown of factors known to regulate XCI, including Yy1 (initiates *Xist*[52] expression), Hnrnpu (tethers *Xist* to the Xi[53]) and Jarid2 (directs polycomb to *Xist*-localised regions[54,55]). Xmas mESCs were transduced with validated shRNAs (Supplementary Fig. 3a) 6 days prior to differentiation, maintaining antibiotic selection to ensure hairpin activity. Flow cytometry revealed knockdown of *Yy1* or *Jarid2* inhibited XCI relative to a non-silencing control (Nons) (Fig. 4c). To overcome the variable

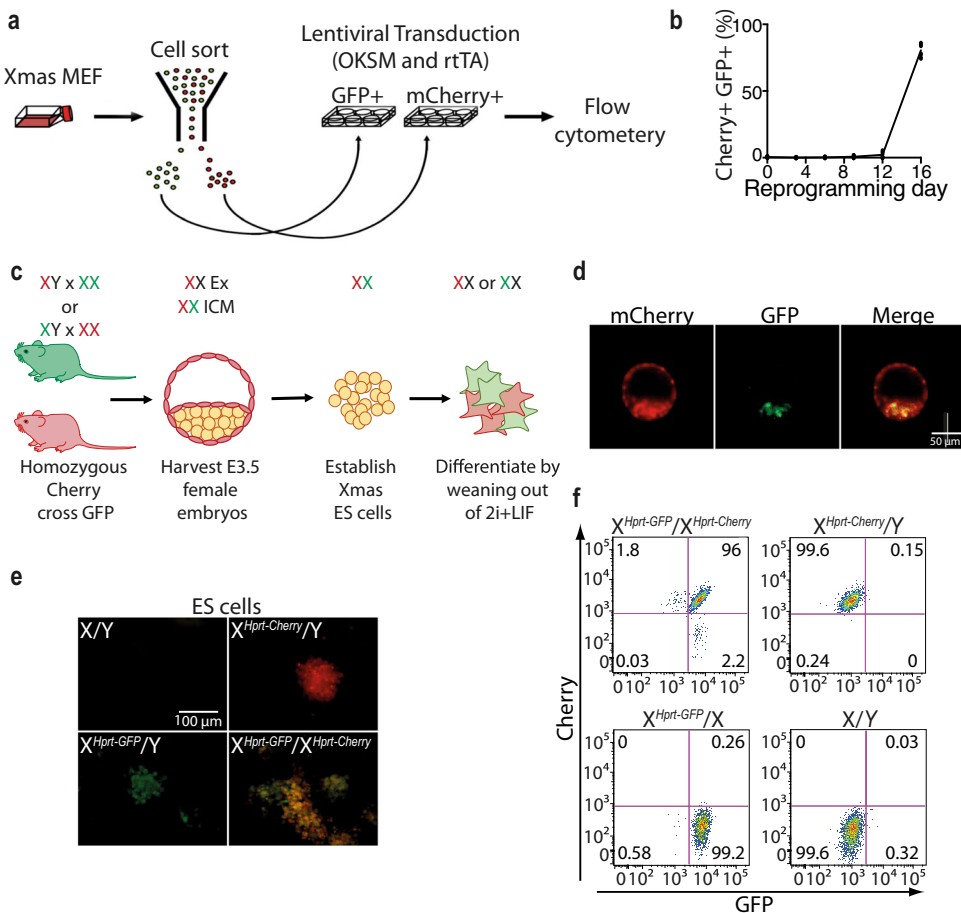

**Fig. 2 Xmas reporters allow detection of X chromosome expression in mESCs and iPSCs. a** Schematic showing the strategy for reprogramming and analysis of Xmas MEFs. **b** Flow cytometry data from primary female Xmas MEFs during the reprogramming process ($n = 4$ independent replicates). **c** Schematic of the breeding strategy to produce Xmas mESCs, their XCI status in vivo and during culture and differentiation in vitro. Extraembryonic (Ex), Inner Cell Mass (ICM). **d** Live fluorescent image of $X^{Hprt-GFP}/X^{Hprt-mCherry}$ Xmas female blastocysts. **e** Live fluorescent image of cultured mESCs from the indicated genotypes carrying different combinations of the fluorescent reporter alleles. Images shown are from a single experiment. **f** Flow cytometry of cultured mESCs from the indicated genotypes carrying different combinations of the fluorescent reporter alleles. Source data are provided as a source data file.

percentage of XO cells in starting populations, each experiment was normalised to the matched Nons control (Fig. 4d). Knockdown of *Hnrnpu* caused rapid loss of double positivity prior to differentiation, likely due to Hnrnpu's role in pluripotency[56]. This suggests Xmas mESCs can identify pluripotency factors. To instead test Hnrnpu's function in XCI, we transduced cells with shRNA at day 2 of differentiation, so knockdown occurs following exit from pluripotency. Using this strategy, Hnrnpu knockdown no longer caused accelerated loss of double positivity, instead inhibiting XCI (Fig. 4e, f). Depletion of Yy1 now no longer delayed XCI, consistent with Yy1's early role in XCI[52,57] prior to knockdown. To confirm these results reflect a direct effect on XCI, rather than altered differentiation kinetics, we performed real-time quantitative PCR (qRT-PCR) for the pluripotency genes Nanog and Sox2 in differentiating Xmas mESCs, finding no difference in expression level following any knockdown apart from the expected accelerated loss of pluripotency upon knockdown of Hnrnpu prior to differentiation (Supplementary Fig. 3b, c). These data indicate by varying the time of shRNA transduction, Xmas mESCs suggest when a factor is required during the process of XCI.

**A targeted shRNA screen in differentiating Xmas mESCs reveals Smarcc1 and Smarca4 are required for XCI.** The tractability of Xmas mESCs, allowed us to screen for genes that establish the Xi. Our previous mouse genetic screen identified epigenetic regulators of transgene variegation[58–66]. This screen yielded a list of seventeen candidate proteins, some of which are also known to be required for XCI[12,23,34,35,67]. Given this, we selected this suite of genes to target in our screen. Xmas mESCs were transduced with validated hairpins (Supplementary Fig. 3a) at day 2 of differentiation and assessed by flow cytometry at day 6; a timepoint we consistently observe effects from gene knockdown (Fig. 4e, f). Strikingly, XCI was impaired by shRNAs against nucleosome remodellers *Smarcc1* and *Smarca4*, both members of the ESC-specific BAF complex[68–70] (Fig. 5a). We validated the screen result in a Xmas mESC differentiation timecourse, detecting the failure of XCI by day 5 for knockdowns of either *Smarcc1* or *Smarca4* (Fig. 5b and Supplementary Fig. 4a). As genes in our screen are knocked down following the induction of differentiation, we cannot exclude roles early in XCI for the genes that did not readout, but here we chose to focus on the role of *Smarcc1* and *Smarca4*.

To determine the extent *Smarcc1* and *Smarca4* knockdown impairs XCI across the whole X chromosome, we performed RNA-seq in differentiating $X^{FVB}X^{CAST}$ F1 mESCs (Fig. 5c and Supplementary Fig. 4c, e). Knockdown was maintained throughout the assay (Supplementary Fig. 4b, d) and resulted in increased gene expression from $X^{FVB}$ (preferential Xi) at day 6 of differentiation at the majority of informative X-linked genes (Fig. 5d, e and

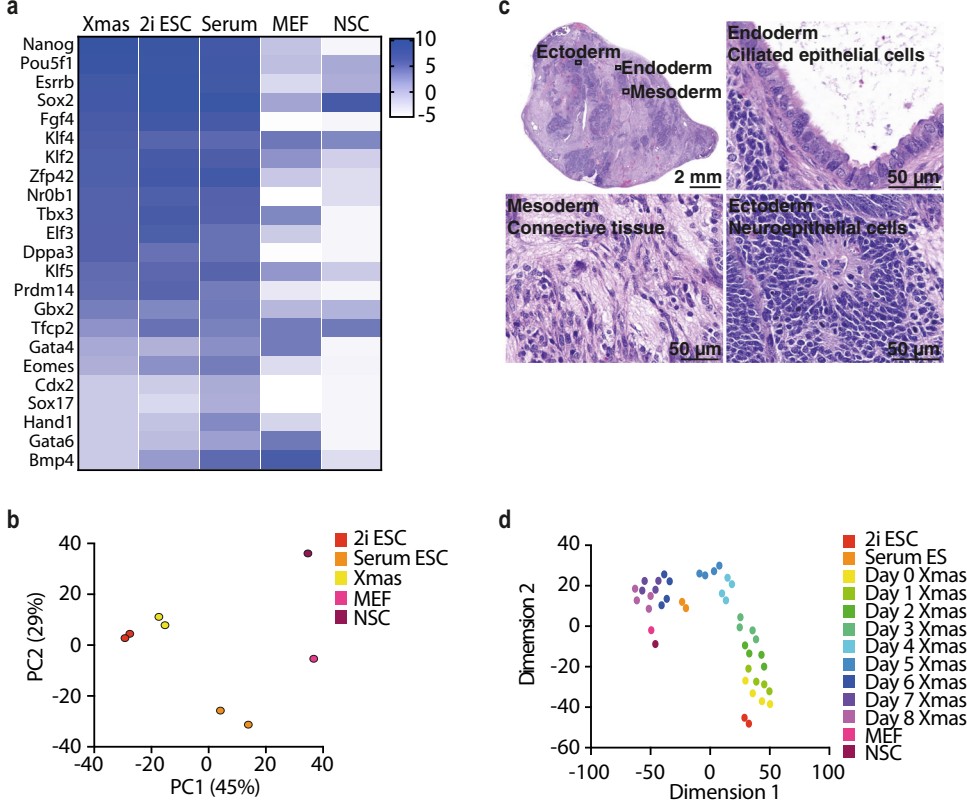

**Fig. 3 Xmas mESCs are pluripotent and differentiate as expected. a** Heatmap showing average expression ($\log_2$ rpm) of pluripotency genes in Xmas mESCs ($n = 2$ independent replicates) and published transcriptomes of mESCs grown in serum or 2i, MEFs or NSCs. **b** Principle component analysis of RNA-seq data from Xmas mESCs compared with published transcriptomes of mESCs grown in serum or 2i, MEFs or neural stem cells (NSCs). **c** Representative images of teratomas produced following injection of Xmas mESCs into nude mice ($n = 4$ independent replicates), with differentiated cell types from endodermal, mesodermal and ectodermal lineages shown. **d** tSNE plot comparing the transcriptomes of Xmas mESCs ($n = 4$ independent replicates) from day 0 to day 8 of differentiation against published transcriptomes of mESCs grown in serum or 2i, MEFs or NSCs.

Supplementary Fig. 4c, e, f and Supplementary Data 2, 3), suggesting both Smarcc1 and Smarca4 are required for chromosome-wide silencing. We next focussed on *Smarcc1*, performing RNA-seq during differentiation and found the persistent failure of XCI in knockdown cells, detectable from day 5 (Fig. 5f, g and Supplementary Fig. 4g).

Despite Smarcc1 and Smarca4 both being members of the same complex, there were no significantly differentially expressed genes in common between *Smarcc1* and *Smarca4* knockdown groups. Similarly, depletion of different subunits of the BAF complex has previously been reported to result in different chromatin states, accessibility and transcription[71,72]. In this context, the lack of overlap suggests the mechanism by which they regulate XCI is not via a secondary gene or delayed differentiation. Indeed, we found no misexpression of known protein regulators of XCI following Smarcc1 or Smarca4 depletion (Fig. 5h and Supplementary Fig. 4h, Supplementary Data 4), nor consistent changes to pluripotency factors or early differentiation genes (Supplementary Fig. 4i, j). To further assess whether *Smarcc1* or *Smarca4* knockdown delayed differentiation, we performed a correlation analysis by calculating the Euclidean distance between genes in our female RNA-seq dataset and a differentiation timecourse in male cells, finding no evidence of delay (Fig. 5i and Supplementary Fig. 4k, l and Supplementary Data 5). Subsequent qRT-PCR experiments for common pluripotency and differentiation genes were also consistent with normal differentiation upon *Smarcc1* or *Smarca4* depletion (Supplementary Fig. 4m). To experimentally separate the XCI role of Smarcc1 and Smarca4 from potential roles in pluripotency, we differentiated Xmas mESCs and

transduced with shRNA at day 3 to achieve depletion later during differentiation, again detecting the failure of XCI (Supplementary Fig. 4n). The timing of knockdown of Smarcc1 and Smarca4 in this and the earlier experiment is after the stage when factors important in the initiation of XCI, such as Yy1, readout in this screen. Together, these data suggest Smarcc1 or Smarca4 depletion do not influence XCI via altered timing of exit from pluripotency or differentiation, but instead may be due to a direct effect on the establishment of XCI.

**Smarcc1 and Smarca4 are required at the establishment of XCI.** We next sought to investigate when the defect in XCI first occurred in the Smarcc1 and Smarca4 knockdown cells, starting with an examination of *Xist* activation and spreading. RNA fluorescence in situ hybridisation (FISH) for *Xist* detected no signal in mESCs and during an mESC differentiation timecourse found no difference in the number of cells with an *Xist* focus following knockdown of Smarcc1 or Smarca4 at any timepoint; however, depleted cells were largely unable to form the distinctive *Xist* cloud apparent at day 6 of differentiation (Fig. 6a, Supplementary Fig. 5a, b). To understand potential defects in *Xist* spreading, we performed a volumetric analysis of *Xist* foci, finding that depletion of *Smarcc1* had no effect on *Xist* spreading at day 4 or 5 of differentiation (Fig. 6b). Interestingly, depletion of *Smarca4* appeared to show accelerated *Xist* spreading at day 4 of differentiation, likely due to the role of Smarca4 in maintaining pluripotency[68–70], however, this acceleration was resolved by day 5 and is followed by a clear failure to coat the Xi at day 6. To gain

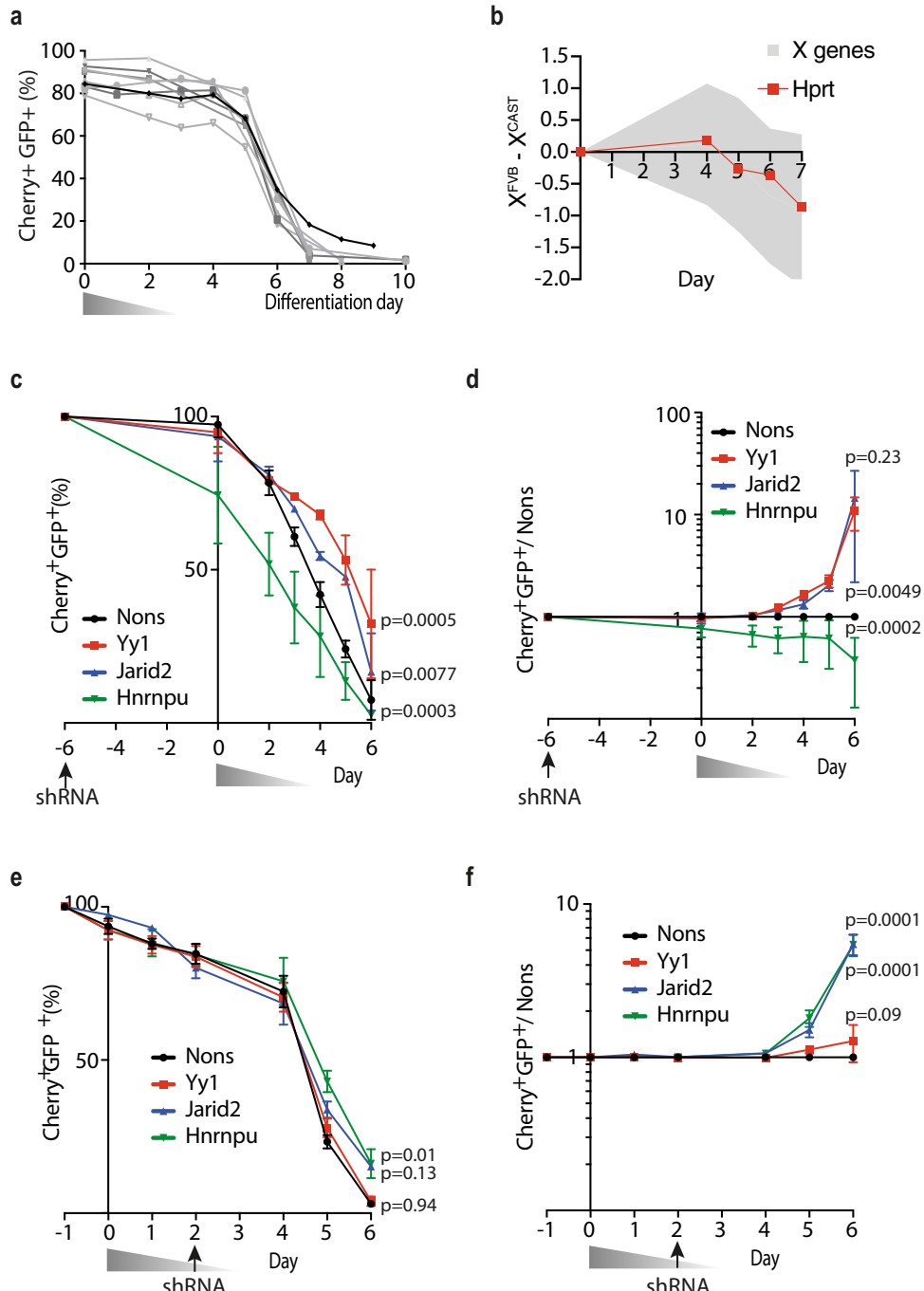

**Fig. 4 Xmas mESCs detect impaired XCI during differentiation. a** Flow cytometry data showing the kinetics of the fluorescent reporter alleles during differentiation and XCI of Xmas mESCs for multiple cell lines ($n = 9$ independent replicates). The triangle represents weaning from 2i to differentiation media in 25% increments over 3 days. **b** RNA-seq timecourse data from differentiating $X^{FVB}X^{CAST}$ mESCs showing the ratio of $X^{FVB}$ gene expression compared with the $X^{CAST}$ ($X^{FVB}—X^{CAST}$ log$_2$). *Hprt* expression is represented as a red line and other X-linked genes as grey shade. **c–f** Flow cytometry data showing the kinetics of the fluorescent reporter allele expression changes during differentiation and XCI of Xmas mESCs. Cells were challenged with shRNAs against the indicated known regulators of XCI or control (Nons) either prior to differentiation depicted as either raw data (**c**) or normalised to Nons (**d**), or during differentiation as either raw data (**e**) or normalised to Nons (**f**). Triangles represent weaning from 2i media into differentiation media and arrows indicate the day of shRNA viral transduction. $n = 3–6$ independent replicates from two independent shRNAs per gene, error bars indicate s.e.m., two-way ANOVA, *p* value is given for the entire timecourse of that gene knockdown. Source data are provided as a source data file.

a more accurate readout of transcript levels and as these FISH experiments do not discriminate between *Xist* and *Tsix* RNA, we performed qRT-PCR, finding *Xist* to be 100-fold more highly expressed than *Tsix*. Together with no signal at day 0 these data indicate the FISH signal is likely *Xist* (Supplementary Fig. 5c, d). No effect on *Tsix* RNA was observed following depletion of either

*Smarcc1* or *Smarca4*, whereas *Xist* RNA levels were slightly decreased.

The change in *Xist* levels and localisation only later during the timecourse of XCI suggest that *Xist* is correctly induced but may be destabilised due to failure to localise to the Xi[73,74]. To test this directly we performed a series of experiments in a male mESC

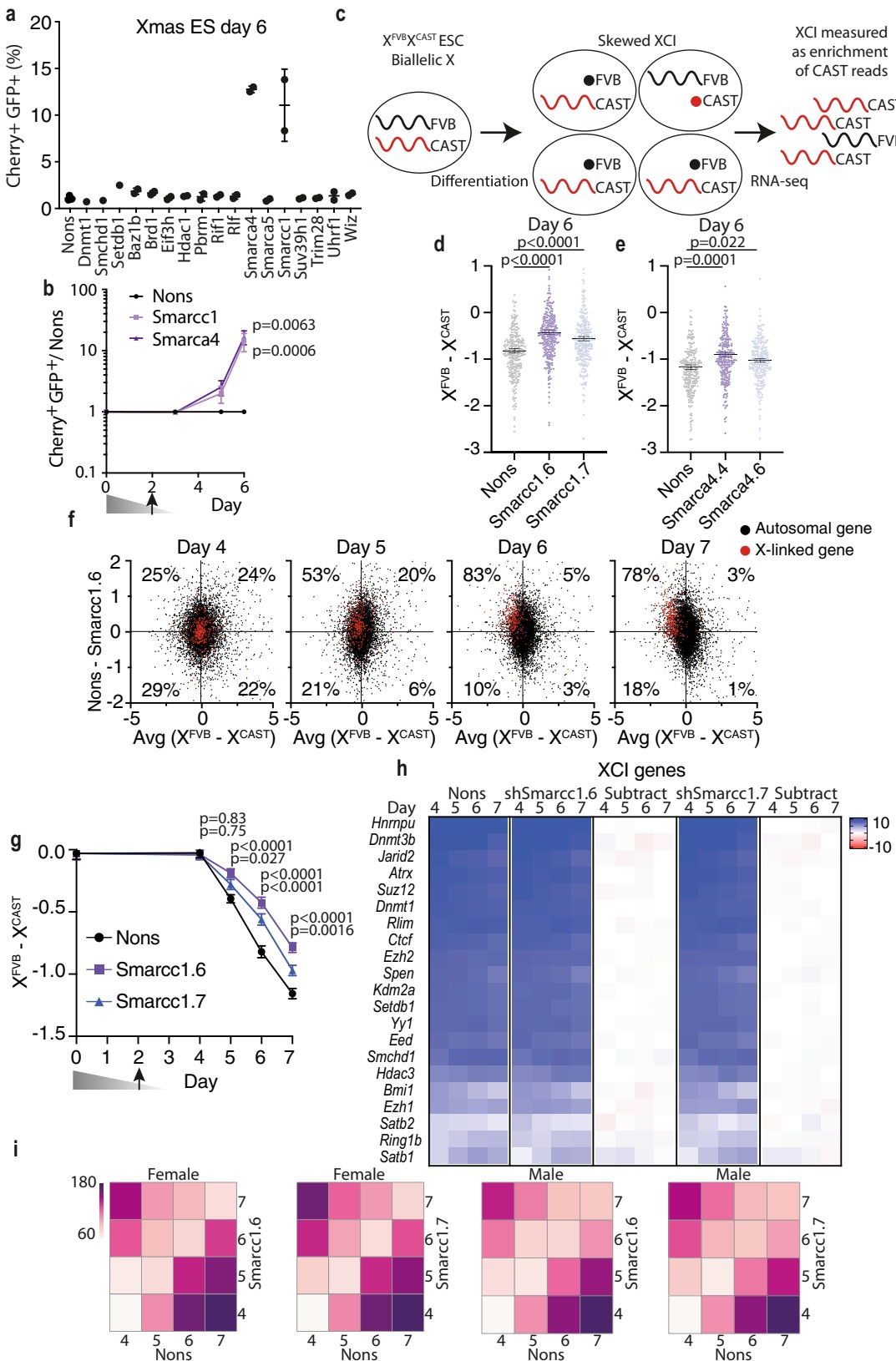

line that carries a doxycycline-inducible *Xist* transgene on chromosome 17[75]. *Xist* induction, spreading and silencing are very rapid in this model, so to test induction of *Xist* we performed *Xist* RNA FISH at 30 min post doxycycline induction, prior to significant spreading of *Xist*, and found no difference in the proportion of nuclei producing an *Xist* signal in controls and

*Smarcc1* or *Smarca4* depleted cells (Supplementary Fig. 5e, f). This timepoint is akin to day 4 of differentiation in Xmas mESCs with endogenous *Xist*. When measured by qRT-PCR, *Xist* levels were normal following 24 h of continual doxycycline driven expression in control and depleted cells (Supplementary Fig. 5g). To determine *Xist*'s ability to spread and establish XCI, we

**Fig. 5 Screen in Xmas mESCs identifies Smarcc1 and Smarca4 as regulators of XCI. a** Flow cytometry data at day 6 of Xmas mESC differentiation following shRNA transduction at day 2 against candidate genes ($n = 2$ hairpins per gene, error bars indicate S.D.). **b** Flow cytometry data normalised to Nons along timecourse of Xmas mESC differentiation following knockdown of Smarcc1, Smarca4 or Nons ($n = 4$ independent replicates with two shRNAs per gene, error bars indicate s.e.m. Two-way ANOVA, $p$ value given). **c** Schematic of skewed XCI during differentiation of $X^{FVB}X^{CAST}$ mESCs. **d, e** Allele-specific RNA-seq data of $X^{FVB}X^{CAST}$ mESCs at day 6 of differentiation following knockdown of Smarcc1 (**d**) and Smarca4 (**e**). Each point represents the $X^{FVB}$-$X^{CAST}$ $\log_2$ expression for informative X-linked genes ($n = 239$–$281$ genes, error bars indicate s.e.m. Two-tailed Student's unpaired $t$ test, $p$ value given). **f** Graphs showing RNA-seq data designed to compare gene expression between X chromosome and autosomes. Each point represents an informative gene, X-linked genes in red, autosomal genes black. The x-axis shows the ratio of expression from FVB to CAST ($X^{FVB}$—$X^{CAST}$ $\log_2$), therefore XCI is observed as a left shift of the red dots along the $x$ axis. The $y$ axis shows ratio of expression from Nons compared with knockdown with Smarcc1.6 (Nons—Smarcc1.6 $\log_2$FC), therefore failure of XCI upon knockdown is observed as an upward shift along the $y$ axis. Black dots give an indication of global trends in autosomal gene expression. Dotted lines indicate medians and percentages show the X-linked genes falling into each quadrant. **g** RNA-seq time course data showing the ratio of $X^{FVB}$ gene expression compared to $X^{CAST}$ ($X^{FVB}$—$X^{CAST}$ $\log_2$). Error bars indicate s.e.m. of informative genes ($n = 246$ – $271$ genes), Two-tailed Student's unpaired $t$ test, $p$ value given. **h** Gene expression ($\log_2$ rpm) of known XCI regulators, the difference between knockdown and control (subtract, Nons—knockdown) indicated. Scale bar represents both $\log_2$ rpm and $\log_2$FC. **i** Heat maps showing Euclidean distance in gene expression ($\log_2$ rpm) between Smarcc1 knockdown and Nons control along a differentiation timecourse of male or female mESCs. Source data are provided as a source data file.

performed a timecourse of immunoflourescence for mCherry (which is tethered to Xist in this cell model), finding these cells were less able to form an Xist cloud and H3K27me3 foci than control cells at days 1 and 2 post Xist induction (Supplementary Fig. 5h, i). These timepoints are akin to day 6 of differentiation in Xmas mESCs. These data suggest that the defect upon Smarcc1 or Smarca4 depletion is downstream of Xist induction. Using the inducible system to study Xist degradation, we found the reduction in Xist transcript is likely due to destabilisation of the transcript following failure to localise to the Xi (Supplementary Fig. 5j). Finally, Smarcc1 or Smarca4 depleted cells displayed a survival advantage over control cells, further supporting failed Xist-induced gene silencing in depleted cells (Supplementary Fig. 5k). Taken together with our prior data showing failure of gene silencing in female cells detectable from day 5 of differentiation, normal Xist induction at days 4 and 5, but inablilty to form an Xist cloud and H3K27me3 foci at day 6, these data suggest that Smarcc1 and Smarca4 are required early in the establishment of silencing on the Xi, beyond which key XCI events fail. Notably, this is a differentiation-free model of Xist-induced silencing and therefore disentangles the roles of Smarcc1 and Smarca4 from any potential role in differentiation.

Finally, to test for potential roles in the maintenance of XCI we performed Smarcc1 and Smarca4 knockdown in post-XCI Xmas MEFs sensitised to X reactivation by treatment with the DNA methyltransferase inhibitor 5-azacytidine. Knockdown of either gene was unable to reactivate the silent reporter allele (Supplementary Fig. 5l), but neither was the known maintenance factor Dnmt1. Reversal of XCI during maintenance is difficult, so we employed a more sensitive system[76–80] where MEFs carry a silent multi-copy GFP transgene on their Xi by virtue of an Xist knockout in trans to the reporter (Xi$^{GFP}$Xa$^{\Delta Xist}$ MEFs)[23,81]. Again, we found no reactivation of the silent reporter upon Smarcc1 or Smarca4 knockdown, despite positive controls producing readily detectable GFP (Supplementary Fig. 5m), therefore providing no evidence for a role in the maintenance of XCI.

**BAF complex localisation to the Xi is dynamic.** To determine whether Smarca4 acts directly on the Xi to establish XCI, we performed immunofluorescence for Smarca4 together with H3K27me3, a marker of a later stage establishing Xi and found colocalization of Smarca4 and H3K27me3 in some cells but exclusion in others at day 6 of differentiation (Supplementary Fig. 6a, b). Moreover, upon either Smarcc1 or Smarca4 depletion fewer cells are able to form H3K27me3 foci at day 6 of differentiation (Supplementary Fig. 6c, d), suggesting XCI is unable to

proceed to this point. As Smarca4 is absent from the Xi in terminally differentiated cells[12,82], these data suggest that Smarca4 is present on the establishing Xi while it is required, but excluded upon completion. With depleted Smarcc1 or Smarca4, gene silencing fails at day 5 and Xist spreading and H3K27me3 deposition fails at day 6, suggesting that Smarcc1 and Smarca4 are active prior to these events. To assess the presence of Smarca4 on the establishing Xi at day 4 of differentiation, prior to H3K27me3 deposition, we performed ChIP-seq for Smarca4 with either a Smarcc1 knockdown or non-targeting control in both male and female mESCs, finding that Smarca4 was indeed enriched on the establishing Xi at this early stage of XCI with an abundance of peaks on the female X that cannot be accounted for simply by the presence of 2 X chromosomes compared with males (Fig. 6c, d). Smarca4 peaks were found at promoters on the X chromosome (and autosomes), some of which are only found in control females (Fig. 6e, f and Supplementary Fig. 6e–g). These data suggest Smarca4 may play a role at promoters on the establishing Xi at day 4 of differentiation. Interestingly, very little Smarca4 ChIP-seq signal was produced in female cells upon Smarcc1 knockdown, including at X-linked promoters, suggesting these proteins are acting together as the BAF complex to establish the Xi.

**The BAF complex depletes nucleosomes at Xi promotors prior to establishment of XCI.** That the BAF complex binds to the Xi during the establishment phase of XCI and contributes functionally to establishment, suggests it acts directly on the Xi as a nucleosome remodeller. Therefore, we profiled nucleosome occupancy in differentiating $X^{FVB}X^{CAST}$ mESCs by allele-specific Nucleosome Occupancy and Methylome Sequencing (NOMe-seq)[83–85]. Nucleosome dynamics during XCI establishment have not been reported previously, so initially we concentrated on normal mESC differentiation (Nons control). The reduced coverage of allele-specific data precluded gene-specific analyses, so we averaged across X-linked genes, finding different nucleosome kinetics between X chromosomes. $X^{CAST}$ (preferential Xa) promotors are slightly open in mESCs, remaining so at day 4 of differentiation before opening further at day 5, then restricting again at day 6 (Fig. 6g, h). Similar kinetics were observed on autosomes (Supplementary Fig. 6h, j). Similar patterns were also recently seen in NOMe-seq data sets from equivalent stages of post-implantation embryos[86], suggesting promoter opening is common during the transition from pluripotency to lineage-restricted states. The $X^{FVB}$ (preferential Xi) followed different kinetics, where promotors were initially slightly open, similarly to $X^{CAST}$, but became nucleosome depleted at day 4, a day earlier

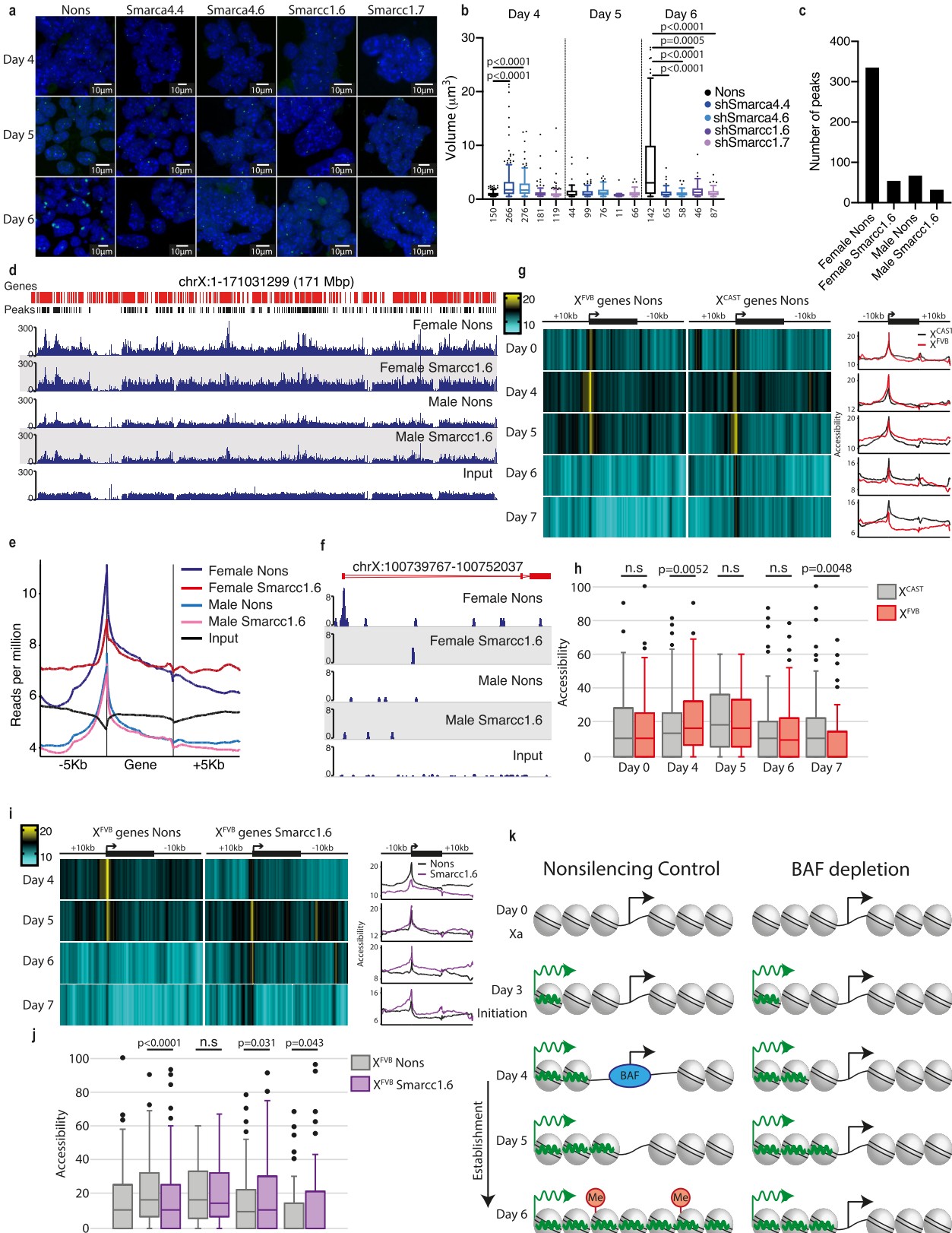

than X$^{CAST}$ (Fig. 6g, h), suggesting Xi promotors become accessible prior to gene silencing at day 5. The X$^{FVB}$ subsequently becomes progressively more nucleosome dense at both promotors and gene bodies, as expected to occur with gene silencing. No allelic differences were observed at autosomal genes (Supplementary Fig. 6h).

To address the functional role of nucleosome depletion prior to silencing, we produced a *Smarcc1* knockdown NOMe-seq time-course. Depleted cells were unable to open X$^{FVB}$ promoters at day 4 and instead followed kinetics similar to X$^{CAST}$, consistent with the XCI failure observed by RNA-seq (Fig. 6i, j). These data suggest an inability to open promotors at day 4 results in failure

**Fig. 6 Smarcc1 opens Xi promotors in order for the establishment of XCI to proceed. a** *Xist* RNA FISH in female mESCs at differentiation days 4, 5, 6 following knockdown with indicated hairpins. *Xist* staining green, DAPI blue. **b** Volume of *Xist* foci in **a**. Foci measured (*n*) are indicated. The line indicates median, box 25th to 75th percentile, error bars 5th to 95th percentile, dots indicate outliers. Two-tailed Student's unpaired *t* test, statistically significant *p* values only given. **c–f** Smarca4 ChIP-seq in male and female mESCs at differentiation day 4, with Smarcc1 knockdown or non-silencing control, **c** number macs2 peaks, **d** coverage plot of X chromosome, **e** average read density at X-linked genes ±5kb and **f** an example coverage plot. **g** Nucleosome occupancy (% GpC methylation) during female mESC differentiation determined by NOMe-seq averaged across genes on the $X^{FVB}$ and $X^{CAST}$, displayed as a heatmap or smoothed histogram. **h** Accessibility of $X^{FVB}$ or $X^{CAST}$ promoters during female mESC differentiation determined by NOMe-seq. Line indicates median, box 25th to 75th percentile, error bars 5th to 95th percentile and dots indicate outliers. *n* = 74 to 261 informative promoters. One-tailed Student's unpaired *t* test without outliers, *p* value is given, non-significance (n.s.). **i** Nucleosome occupancy (% GpC methylation) during female mESC differentiation determined by NOMe-seq averaged across all genes on $X^{FVB}$ upon *Smarcc1* knockdown, displayed as a heatmap or smoothed histogram. **j** Accessibility of $X^{FVB}$ promoters upon *Smarcc1* knockdown during female mESC differentiation determined by NOMe-seq. Line indicates the median, box 25th to 75th percentile, error bars 5th to 95th percentile and dots indicate outliers. *n* = 58 to 261 informative promoters. One-tailed Student's unpaired *t* test without outliers, *p* value is given, non-significance (n.s.). **k** Model for BAF regulation of establishment of XCI. Open and closed chromatin depicted by nucleosome spacing, green lines represent *Xist*, black arrows transcription and red paddles marked 'Me' H3K27me3. This figure depicts timing during differentiation when key XCI events occur; Xist induction (day 3), BAF occupancy at promoters and promoter opening (day 4), failure of gene silencing (day 5) and failure of *Xist* cloud formation and H3K27me3 deposition (day 6). Source data are provided as a source data file.

to establish the Xi and, together with our ChIP-seq data showing localisation of Smarca4 to promoters at this time, this appears to be directly mediated by the BAF complex. No effect of *Smarcc1* depletion was observed on the $X^{CAST}$ or autosomes (Supplementary Fig. 6i, j), however, there are likely gene-specific abnormalities not detected, and potentially cell-type-specific effects that would not be revealed by our undirected differentiation method. NOMe-seq also detects methyl-cytosine and showed mESCs were globally hypomethylated, remaining so at promotors during differentiation, but becoming increasingly methylated at intergenic regions and gene bodies. Methylation of CpG islands on the Xi is a feature of XCI maintenance. As expected, given the timing of our samples, we did not observe such methylation occurring, and no difference was observed between the $X^{FVB}$ and $X^{CAST}$ nor upon *Smarcc1* depletion (Supplementary Fig. 6k, l).

To validate our NOMe-seq data, we performed MARs-qPCR, a micrococcal nuclease-based method to assess site-specific nucleosome occupancy[87] in differentiating Xmas mESCs for a subset of Smarcc1-responsive or unresponsive promoters. Note that increased MARS-qPCR signal indicates decreased accessibility and is therefore directionally inverse to NoME-seq signal. In agreement with NOMe-seq data, we found that all but one of the Smarcc1-responsive promoters were also less open upon *Smarcc1* or *Smarca4* depletion at day 4 of differentiation, whereas at day 6 they were more open, indicative of failed gene silencing (Supplementary Fig. 6m). Smarcc1-unresponsive promoters showed no effect. In this assay we also included the *Xist* and *Tsix* promoters. In agreement with our previous data suggesting Smarcc1 or Smarca4 are not required for *Xist* activation, we found no change in nucleosome occupancy at the *Xist* promoter upon depletion of *Smarcc1* or *Smarca4* at day 4 of differentiation. The *Tsix* promoter, however, was nucleosome enriched at day 4 of differentiation upon *Smarcc1* or *Smarca4* depletion, likely reflecting the need to silence *Tsix* on the Xi and suggesting that, similar to other genes on the Xi, promoter relaxation is required for this process. Interestingly, despite altered nucleosome occupancy *Tsix* expression is not affected by BAF depletion at the time points measured (Supplementary Fig. 5d).

## Discussion

We wished to use XCI in differentiating female mESCs as a model epigenetic system where we could learn more about features of epigenetic silencing. Despite being of high interest, complications with their in vitro culture have meant female mESCs are experimentally underutilised. To allow us to study XCI in the native female context, we created Xmas mESCs as a tractable

fluorescent reporter system that requires minimal manipulation of these delicate cells. Xmas reporters enabled rapid and regular monitoring of XX vs XO cells in culture so that we could ensure a highly XX population of pluripotent female cells. The fluorescent reporters then also allowed us to monitor X inactivation during differentiation, to perform a screen for regulators of XCI establishment during normal female mESC differentiation. All previous screens for XCI regulators were performed either in differentiated cells for factors that maintain XCI[12,23,35,88–93], or non-native systems that instead induce *Xist* out of context (from an autosome in male cells or prior to exit from pluripotency in female mESCs[10,11,75,94]). These screens have been highly fruitful but will miss genes required only for the establishment of XCI or that require a differentiating cell state to be active. Although small in scale, our screen suggests Xmas mESCs will be suitable for high-throughput screening approaches.

The screen revealed a role for Smarcc1 and Smarca4 in the establishment of epigenetic silencing of the X chromosome. Smarcc1 (also known as Baf155) and Smarca4 (also known as Brg1) are members of the chromatin remodelling BAF complex, with Smarcc1 being the core subunit around which the complex forms[95] and Smarca4 one of a variable number of catalytic ATPase subunits[96,97]. The BAF complex contains different subunits dependent on cell type, with Smarcc1 and Smarca4 members of an mESC-specific complex, known as esBAF[70]. When mESCs are depleted of either Smarcc1 or Smarca4 they display reduced expression of core-pluripotency transcription factors, reduced self-renewal, and loss of pluripotency[68–70]. Here we reveal a role for esBAF during exit from pluripotency in females, with *Smarcc1* or *Smarca4* depletion causing failure of XCI. Deletion of *Smarcc1* or *Smarca4* in mice is lethal peri-implantation, and although consistent with XCI failure, male embryos also die, precluding conclusions about their XCI roles in vivo[98–100]. Two prior screens for regulators of establishment of XCI did not identify members of the BAF complex, however, these were performed in pluripotent mESCs with inducible *Xist* and so were unlikely to identify genes with dual roles in XCI and pluripotency, such as *Smarcc1* and *Smarca4*[75,94].

A previous study found Smarca4 was required for maintenance of XCI in a somatic cell line[12], however, a later study by the same group reported that *Xist* repels Smarca4 from the Xi in order to maintain silencing[82], a somewhat contradictory finding. Here we also find no evidence for maintenance of XCI by the esBAF complex. Instead, the clear failure to establish silencing following *Smarcc1* depletion inspired us to profile nucleosome occupancy during establishment of XCI. This time course revealed that Xi promotors become nucleosome depleted at the very earliest stages

of gene silencing. Importantly, we functionally link promoter opening to gene silencing; cells with depleted *Smarcc1* fail to open promotors and fail to establish the Xi, with the resulting Xi following a similar trajectory to the Xa, both in terms of nucleosome positioning and gene silencing. Our data suggest a model where esBAF is recruited to the future Xi to make it accessible, perhaps to silencing factors or to enable *Xist* spreading, with the BAF complex subsequently excluded once XCI is complete (Fig. 6k). Therefore, Smarcc1 creates a chromatin state that allows establishment of silencing to proceed.

The timing at which we observe key silencing events is pertinent. *Xist* is induced and its spreading appears normal at day 4 of differentiation. Smarca4 is present at promoters of the Xi at day 4 and Smarcc1-mediated promoter opening occurs the same day, placing nucleosome depletion at promoters early in the ontogeny of epigenetic silencing. Other key events are downstream of promoter opening, occurring on subsequent days. Upon knockdown of *Smarcc1* or *Smarca4*, we detect the failure of gene silencing from day 5 of differentiation, the first day X-linked gene silencing is measurable. On day 6, there is an observable failure to form the distinctive *Xist* cloud in *Smarcc1* and *Smarca4* depleted cells and H3K27me3 deposition fails (Fig. 6k). This suggests an inability of *Xist* to spread or localise to the Xi late in the establishment of XCI, and the timing implies this is a consequence of failure to establish the Xi, rather than a direct requirement of nucleosome remodelling for *Xist* spreading. It is important to note that we cannot exclude further roles for the BAF complex either in the induction of *Xist* or in facets of establishment and maintenance of XCI that are untested here. We have not determined the mechanism by which Smarcc1 or Smarca4 are recruited to the establishing Xi, however, we do not believe this is likely to be through direct interaction with *Xist*. Firstly, Smarcc1 and Smarca4 do not possess classic RNA binding domains. Secondly, although a previous study found *Smarca4* bound to *Xist* in vitro in differentiated cells, follow-up work by the same group showed *Xist* repelled Smarca4[12,82], and other surveys of *Xist* interactors found no evidence of direct Smarca4 or Smarcc1 binding in cells relevant to the establishment of XCI[10,11,101]. A recent paper intriguingly found Spen was required early for the establishment of XCI and localised to promotors[102], raising the possibility that Smarcc1 and Smarca4 may be recruited by Spen.

In summary, Xmas mESCs enabled the discovery of previously unknown requirements for establishment of the Xi, namely nucleosome remodeller-dependent chromatin opening, that occurs prior to gene silencing. It remains unclear whether this is also a requirement for autosomal gene silencing, however, as all aspects of XCI gene silencing are also features of epigenetic silencing more broadly, this is a likely possibility. The Xmas mESC system provides a renewable resource of high-quality female mESCs and makes the study of XCI and other aspects of female-specific pluripotency more feasible than ever before.

## Methods

**Animal strains and husbandry**. Animals were housed and treated according to Walter and Eliza Hall Institute (WEHI) Animal Ethics Committee approved protocols (2014.034, 2018.004). Temperature maintained between 19 °C and 24 °C, 40–60% humidity, and a light/dark cycle of 14 h/10 h. Xmas mice are C57BL/6 background and were maintained as homozygous lines. D4/XEGFP mice were obtained from Jackson Labs and backcrossed onto the C57BL/6 background. Xist[ΔA] mice[81] were obtained from Dr. Graham Kay, Queensland Institute of Medical Research, and kept on a 129 background. Castaneus (CAST/EiJ) mice were obtained from Jackson Labs and maintained at WEHI. FVB/NJ mice were obtained from stocks held at WEHI. Oligonucleotides used for genotyping are provided in Supplementary Data 6.

**Creation of *Hprt* knock-in alleles**. The *Hprt* targeted alleles were generated by recombination in Bruce4 C57BL/6 mESCs. The targeting construct was produced by recombineering. This construct was designed to introduce an IRES-mCherry-

polyA site or an IRES-eGFP-polyA site sequence 20 bp into the 3′-untranslated region (UTR) of *Hprt*, followed by a PGK-neomycin selection cassette flanked by Frt sites. Note, the mCherry used in the construct contained a synonymous mutation to remove the internal NcoI site. The targeting construct also introduced specific sites useful for the Southern blotting strategy used to validate recombination in targeted mESC clones. These sites were SphI and EcoRV at the 5′-end, after 20 bp of the 3′-UTR before the IRES, and EcoRV and NsiI at the 3′-end before the remainder of the 3′UTR.

Neomycin-resistant clones were screened by Southern blot for their 5′ and 3′ integration sites using PCR amplified probes. The 5′ probe was amplified with the 5′-AAACACACACACACACTCCACAAA-3′ and 5′-GCACCCATTATGCCCTAGA TT-3′ oligos, the 3′ probe was amplified with 5′-GCTGCCTAAGAATGTGTTG CT-3′ and 5′-AAGCCTGGTTTTGGTAGCAG-3′ oligos. Each was cloned into the TopoTA vector. For the Southern blot, DNA was digested individually with EcoRV and SphI. The wild-type allele generated a 17.4 kb band with EcoRV digestion and the 5′ or 3′ probe, and a 9.2 kb and 8.3 kb knockin band for the 5′ and 3′ probe, respectively. The wild-type allele generated a 7.6 kb probe with SphI digestion and the 5′ probe, compared with a 6.4 kb knockin band. The wild-type allele generated an 8.2 kb band with NsiI digestion and the 3′ probe, compared with a 6.7 kb knock-in allele.

One Hprt-IRES-mCherry-pA-Frt-neo-Frt and one Hprt-IRES-eGFP-pA-Frt-neo-Frt correctly targeted clone was selected and used for blastocyst injection. The PGK-neo selection cassette was subsequently removed by crossing to the Rosa26-Flpe deleter strain[103]. The Hprt-IRES-mCherry and Hprt-IRES-GFP alleles were homozygous and maintained on a pure C57BL/6 background. Genotyping of mice was performed by PCR reaction using GoTaq Green Mix (Promega) and 0.5 µM of each primer, as given in Supplementary Data 6.

**Derivation of mESCs**. Female mice were super-ovulated by injecting 5 IU folligon (MSD Animal Health Australia) two days prior, and 5 IU chorulon (MSD Animal Health Australia) on the day of mating with a stud of the opposite genotype. At E3.5, dams were sacrificed, uteri removed and blastocysts flushed from the uterine horns with M2 medium (Sigma-Aldrich). Blastocysts were washed in M2 medium twice, and 2i+LIF medium [KnockOut DMEM (Life Technologies), 1× Glutamax (Life Technologies), 1× MEM non-essential amino acids (Life Technologies), 1× N2 Supplement (Life Technologies), 1× B27 Supplement (Life Technologies), 1× Beta-mercaptoethanol (Life Technologies), 100 U/mL penicillin/100 µg/mL Streptomycin (Life Technologies), 10 µg/mL Piperacillin (Sigma-Aldrich), 10 µg/mL Ciprofloxacin (Sigma-Aldrich), 25 µg/mL Fluconazol (Selleckchem), 1000 U/mL ESGRO Leukaemia Inhibitory Factor (Merck), 1 µM StemMACS PD0325901 (Miltenyi Biotech), 3 µM StemMACS CHIR99021 (Mitenyi Biotech)] twice. Blastocysts were plated in non-tissue culture-treated 24-well plates in 2i+LIF medium. Following 7 days in the culture at 37 °C in a humidified atmosphere with 5% (v/v) carbon dioxide and 5% (v/v) oxygen, outgrowths were moved by mouth-pipetting through trypsin-ethylenediamine tetraacetic acid (EDTA) for 2 min, mESC wash media [KnockOut DMEM (Life Technologies), 10% KnockOut Serum Replacement (Life Technologies), 100 IU/mL penicillin/100 µg/mL streptomycin (Life Technologies)], and finally 2i+LIF. Outgrowths were disrupted by pipetting and transferred into a 24-well plate to be cultured as mESC lines.

**Culture method for mESCs**. ESCs were maintained in suspension culture in 2i+LIF medium on non-tissue culture-treated plates at 37 °C in a humidified atmosphere with 5% (v/v) carbon dioxide and 5% (v/v) oxygen. mESCs were passaged daily by collecting colonies and allowing them to settle in a tube for <5 min. The supernatant containing cellular debris was removed and mESC colonies were resuspended in Accutase (Sigma-Aldrich) and incubated at 37 °C for 5 min to achieve a single-cell suspension. At least 4× volumes of mESC wash media were added to the suspension and cells were pelleted by centrifugation at $600 \times g$ for 5 min, before plating in an appropriately sized non-tissue culture-treated plate, never flasks, in an excess of 2i+LIF media. Cells were assessed for XX karyotype regularly by flow cytometry.

**Differentiation of mESCs**. At least 2 days prior to inducing differentiation mESCs in suspension were allowed to attach by plating onto tissue culture-treated plates coated with 0.1% gelatin. Differentiation was induced by transitioning cells from 2i+LIF media into DME HiHi media [DMEM, 500 mg/L glucose, 4 mM L-glutamine, 110 mg/L sodium pyruvate, 15% fetal bovine serum, 100 U/mL penicillin, 100 µg/mL streptomycin, 0.1 mM non-essential amino acids and 50 µM β-mercaptoethanol] in 25% increments every 24 h. During this time cells were passaged as required. On the day of transferring into 100% DME HiHi media, ~$10^4$ cells per cm$^2$ were plated onto tissue culture-treated plates coated with 0.1% gelatin. Cells were not passaged for the remainder of an experiment and media was changed as required.

**Transduction of mESCs**. Retrovirus was produced as described[33,104] and concentrated by precipitation with 4% PEG 8000 followed by centrifugation. mESCs were either seeded at $10^5$ cells per cm$^2$ on plates that had been coated with 0.1% gelatin, or at ~$10^5$ cells per mL in suspension in 2i+LIF medium containing PEG concentrated viral supernatant and 8 µg/mL polybrene. The next day medium was changed, and cells were selected with 1 µg/mL puromycin. shRNA sequences are

given in Supplementary Data 6. Some of the shRNAs were validated in previous studies[23,105–107].

**Teratoma formation**. Xmas mESCs were pelleted and washed with phosphate-buffered saline (PBS) before passing through a 70 μm cell strainer. In all, $10^5$ cells were resuspended in 200 μl of 50% matrigel (Corning) in PBS and injected subcutaneous into either the left or right flank of CBA/nude mice. Teratomas were harvested after ~60 days, fixed with formalin, embedded in paraffin, and stained with Haemotoxylin and Eosin.

**Derivation and culture of MEFs**. MEFs were derived from E13.5 embryos and cultured in Dulbecco's Modified Eagle Medium supplemented with 10% (v/v) fetal bovine serum at 37 °C in a humidified atmosphere with 5% (v/v) carbon dioxide and 5% (v/v) oxygen.

**Xist:BglSL-mCherry male mESCs**. Male mESCs with inducible Xist:BglSL-mCherry[75] were maintained in KnockOut DMEM (Life Technologies), 1× Glutamax (Life Technologies), 1× MEM Non-Essential Amino Acids (Life Technologies), 9% KnockOut serum replacement (Life Technologies), 100 U/mL Penicillin/100 μg/mL Streptomycin (Life Technologies), 10 μg/mL Piperacillin (Sigma-Aldrich), 10 μg/mL Ciprofloxacin (Sigma-Aldrich), 25 μg/mL Fluconazol (Sell-eckchem), 50 μM β-mercaptoethanol and 1000 U/mL ESGRO Leukaemia Inhibitory Factor (Merck), at 37 °C in a humidified atmosphere with 5% (v/v) carbon dioxide. Xist expression was induced by the addition of doxycycline at 2 μg/mL. Transduction and antibiotic selection were performed for WT mESCs. Xist transcript stability assays were performed by doxycycline induction of Xist for 24 h, followed by a washout of doxycycline for a further 24 h, RNA-extraction and qRT-PCR with primers listed in Supplementary Data 6. Immunostaining of Xist is achieved in these cells using an antibody against mCherry (1:100 NBP2-25158, Novus Biologicals). Cell survival assays were performed by flow cytometry of cells capable of maintaining mCherry expression. These cells were a kind gift from the Neil Brockdorff laboratory.

**qRT-PCR**. Knockdown efficiency of shRNA retroviral constructs was determined using Roche Universal Probe Library (UPL) assays. Relative mRNA expression levels were determined using the $2^{-ddCt}$ method, with Hmbs as a house-keeping control. Probe numbers and oligonucleotide sequences are provided in Supplementary Data 6. qRT-PCR specific for Xist and Tsix was performed as described[108].

**FACS analysis and sorting**. Cells were prepared in KDS-BSS with 2% (v/v) fetal bovine serum, with dead cells and doublets excluded by size and analysed using a BD LSRFortessa cell analyser. Cells were prepared similarly for sorting using a FACSAria. Flow cytometry data were analysed using FlowJo.

Hematopoietic stem and progenitor cells (LSK: Lineage$^-$ Sca1$^+$ c-Kit$^+$ cells) were isolated from fetal livers from E14.5 Xmas female embryos, essentially as described[106]. Dissociated fetal liver cells were incubated with rat monoclonal anti-Ter119 antibody (TER119, produced in house, 1:100 dilution), then mixed with BioMag goat-rat IgG beads (Qiagen) and Ter119$^+$ cells were depleted using a Dynal magnet (Invitrogen). The remaining cells were stained with Alexa700-conjugated antibodies against lineage markers Ter119 (TER119, produced in house, 1:100 dilution), B220 (RA3-6B2, produced in house, 1:100 dilution), CD19 (1D3, produced in house, 1:100 dilution), Gr1 (RB6-8C5, produced in house, 1:100 dilution), CD2 (Rm2.1, produced in house, 1:100 dilution), CD3 (KT3.1.1, produced in house, 1:100 dilution) and CD8 (53-6.7, produced in house, 1:100 dilution), APC-conjugated anti-c-kit/CD117 (ACK4, produced in house, 1:100 dilution) and PE-Cy7-conjugated anti-Sca1 (Ly6A/E, BD Biosciences, 1:100 dilution). Cells were stained with FluoroGold to assess the viability and analysed on a BD LSRFortessa cell analyser.

**X reactivation assay**. Xmas or Xi$^{GFP}$Xa$^{ΔXist}$ MEFs were transduced with shRNA retroviruses, selected with 3–5 μg/mL puromycin, then treated with 10 μM 5-azacytidine 3 days post transduction. Cells were analysed by FACS 7 days post transduction. This assay was run exactly as previously described[23].

**iPSC generation**. Xmas MEFs were cultured and maintained as previously described[109]. Two days before reprogramming, MEFs were dissociated with 0.25% Trypsin-EDTA (Gibco, 25200114) and labelled[109] with anti-mouse BUV395 Thy1.2 (BD Biosciences, 565257; 1:200), anti-mouse BV421 EpCAM (BD Biosciences, 563214; 1:100) anti-mouse, SSEA-1-Biotin (eBioscience, 13-8813-82; 1:400), Streptavidin Pe-Cy7 (BD Biosciences, 557598; 1:200) and DRAQ7 viability dye (Biolegend, 424001). Using a BD Influx cell sorter (BD Biosciences) setup, GFP$^+$/mCherry$^-$/Thy1$^+$/SSEA-1$^-$/EpCAM$^-$ cells and GFP$^-$/mCherry$^+$/Thy1$^+$/SSEA$^-$1$^-$/EpCAM$^-$ cells were isolated and seeded onto 0.1% gelatin-coated six-well plates at $2 \times 10^3$ cells per cm$^2$. On day −1, Doxycycline-inducible OKSM virus (Millipore, SCR512) and m2rtTA virus (Cyagen Biosciences) were added at a multiplicity of infection of two cells in MEF medium supplemented with 2 μg/μL Polybrene (Millipore, TR-1003-G). Plates were immediately centrifuged at $750 \times g$ for 60 min at room temperature and then incubated at 37 °C and 5% CO$_2$. On day 0, medium was removed and supplemented with

mouse iPSC medium[109] containing 2 μg/mL Doxycycline (DOX) (Sigma-Aldrich, D9891). Medium was changed every 2 days for 12 days. After day 12 of reprogramming, DOX was withdrawn from the culture medium. Cultures were subsequently maintained and passaged regularly with mouse iPSC medium. Cells from reprogramming were harvested on days 3, 6, 9, 12 during reprogramming and iPSC passage 1 (day 16+) for flow cytometry analysis. These cells were labelled with anti-mouse BUV395 Thy1.2 (BD Biosciences, 565257; 1:200), anti-mouse BV421 EpCAM (BD Biosciences, 563214; 1:100) anti-mouse, SSEA-1-Biotin (eBioscience, 13-8813-82; 1:400), Streptavidin Pe-Cy7 (BD Biosciences, 557598; 1:200) and DRAQ7 viability dye (Biolegend, 424001). Samples were then analysed by flow cytometry[110]. For each time point, we quantified the percentage of GFP and mCherry positive cells in the populations that were actively undergoing reprogramming by gating in on the timepoints' respective reprogramming intermediates as defined in[109].

**MARS-qPCR**. MARS-qPCR was performed as described[87] on Xmas mESCs differentiated for either 4 or 6 days. Primers used for qPCR are listed in Supplementary Data 6. Relative DNA abundance was determined using the $2^{-ddCt}$ method, with an intergenic region on chromosome 9 used as a control for normalisation.

**RNA-seq library generation and analysis**. For the RNA-seq depicted in Fig. 3a, b, Xmas mESCs were derived and cultured as described above and compared with published data sets[49,50]. For the RNA-seq depicted in Fig. 3d, Xmas mESC lines were derived and differentiated using the methods described here, with samples collected daily for 8 days of differentiation and compared to published datasets[49,50]. For all Smarcc1 and Smarca4 knockdown RNA-seq in female mESCs (Fig. 5), we derived female mESCs by crossing FVB/NJ (FVB) dams with CAST/EiJ (CAST) sires. The resultant female mESC lines were expanded and then differentiated using our culture conditions. We favour this model of XCI which utilises a natural skewing in XCI, over models of non-random XCI forced by genetic deletion as we find these models lead to accelerated and non-random XO karyotypes that produce artefactual results in our hands. Cells were transduced with the indicated shRNAs at day 2 of differentiation and samples taken for RNA-seq at the indicated timepoints. For Smarcc1 and Smarca4 knockdown RNA-seq in male mESCs (Supplementary Fig. 4l), we derived male C57/Bl6 mESCs and expanded and then differentiated them using our culture conditions. Again, cells were transduced with the indicated shRNAs at day 2 of differentiation, and samples were taken for RNA-seq at the indicated timepoints.

For all RNA-seq experiments, cells were harvested from plates by the addition of lysis buffer and RNA extracted with a Quick-RNA MiniPrep kit (Zymo Research). Sequencing libraries were prepared using the TruSeq RNA sample preparation kit (Illumina) and sequenced in-house on the Illumina NextSeq500 platform with 75 bp reads. For non-allele-specific RNA-seq (C57/Bl6 samples), single-end sequencing was performed. Quality control and adapter trimming were performed with fastqc and trim_galore[111], respectively. Reads were aligned to the mm10 reference genome using either tophat[112] or histat2[113]. Expression values in reads per million (RPM) were determined using the Seqmonk package (www.bioinformatics.babraham.ac.uk/projects/seqmonk/), using the RNA-seq Quantitation Pipeline. Further data interrogation was performed using Seqmonk.

For allele-specific RNA-seq (FVBxCAST samples), paired-end sequencing was performed to improve haplotyping efficiency. Quality control and adapter trimming were performed with fastqc and trim_galore[111], respectively. Reads were aligned to a version of mm10 with SNPs between FVB/NJ with CAST/EiJ n-masked, created using SNPsplit[114], using either tophat[112] or histat2[113]. Reads were haplotype phased using SNPsplit[114] and expression values in RPM determined using the Seqmonk package (www.bioinformatics.babraham.ac.uk/projects/seqmonk/), using the RNA-seq Quantitation Pipeline. For X chromosome-specific analysis, genes were determined to be informative when they had at least 50 mapped and haplotyped reads. Further data interrogation was performed using Seqmonk.

Gene set testing and differential gene expression analysis of male mESC was performed by making two groups by pooling samples at all passages from either the traditional culture method or our improved method. Differential expression analysis between the two mESC culture methods was performed on gene-level counts with TMM normalisation, filtering out genes expressed in fewer than half of the samples, using edgeR v3.26.7[115,116]. Model-fitting was performed with voom v3.40.6[117] and linear modelling followed by empirical Bayes moderation using default settings. Differential expression results from voom were used for gene set testing with EGSEA v1.12.0[118] against the c5 Gene Ontology annotation retrieved from MSigDB, aggregating the results of all base methods but 'fry' and sorting by median rank.

Distance matrices of differentiating mESCs were determined between gene expression profiles of either Smarca4 or Smarcc1 knockdown and the Nons control by calculating the Euclidean distance between log$_2$ rpms with the dist function in R v3.6.1.

**ChIP-seq library generation and analysis**. ChIP-seq libraries were prepared from Xmas mESCs at differentiation day 4 using the ChIP-IT High Sensitivity kit (Active

Motif) according to the manufacturer's instructions and 10 µL of antibody against Smarca4 (D1Q7F, Cell Signalling). Sequencing libraries were prepared using the TruSeq DNA sample preparation kit (Illumina) and sequenced in-house on the Illumina NextSeq500 platform with 75 bp single-end reads. Quality control and adapter trimming were performed with fastqc and trim_galore[111], respectively. Reads were aligned to the mm10 reference genome using bowtie2[119]. Duplicate read removal, peak calling and metagene analysis were performed using the Seqmonk package (www.bioinformatics.babraham.ac.uk/projects/seqmonk/).

**Immunofluorescence**. Immunofluorescence was performed as described in ref. [120], with modifications on differentiating Xmas female mESCs at days 5 or 6. Cells were fixed with 3% (w/v) paraformaldehyde in PBS for 10 min at room temperature, washed three times in PBS for 5 min each and permeabilised in 0.5% (v/v) triton X-100 for 5 min. Cells were blocked in 1% (w/v) bovine serum albumin (BSA) in PBS for 20 min, then incubated in primary antibody in the 1% (w/v) BSA overnight at 4 °C in a humid chamber. Primary antibodies used were Smarca4 (1:100 ab110641, Abcam), Smarcc1 (1:100 #11956 S, Cell Signaling), H3K27me3 (1:100 07-449, Millipore or 1:100 C36B11, Cell Signalling Technology) and mCherry (1:100 NBP2-25158, Novus Biologicals). Cells were washed three times in PBS for 5 min each and then incubated with a secondary antibody diluted in 1% (w/v) BSA for 40 min at room temperature in a dark, humidified chamber. Secondary antibodies used were Donkey anti-rabbit IgG Alexa Fluor 555 conjugate (1:500, A315 Thermo Fisher) and Goat anti-rabbit IgG Alexa Fluor 647 conjugate (1:500, A21244 Thermo Fisher). For the simultaneous staining of Smarca4 and H3K27me3, H3K27me3 (C36B11) rabbit mAb Alexa fluor 647 conjugate (Cell Signalling Technology) was used after the secondary antibody was washed off and incubated for 1 hour in a dark humidified chamber at room temperature. Nuclei were stained with DAPI (0.2 µg/mL) in PBS for 5 min at room temperature. Cells were mounted in Vectashield antifade mounting medium (Vector Laboratories) and visualised on LSM 880 or LSM 980 microscopes (Zeiss). For overlap analyses, image analysis was performed in a semi-automated fashion using a custom written Fiji[121] macro, available here https://github.com/DrLachie/smchd1_coloc. Manual segmentation of cells of interest using the region manager. Auto-thresholding methods were used to segment the nuclei and the H3K27me3 region, and the mean intensity of Smarca4 was measured in both the whole nucleus and region containing H3K27me3.

**Xist RNA FISH**. Xist RNA FISH was performed as previously described[105,120] on day 4 or day 5 in differentiated Xmas mESCs. Xist RNA was detected with a 15 kb cDNA, pCMV-Xist-PA, as previously described[122]. The Xist probe was labelled with Green-dUTP (02N32-050, Abbott) by nick translation (07J00-001, Abbott). The cells were mounted in Vectashield antifade mounting medium (Vector Laboratories) and visualised on LSM 880 or LSM 980 microscopes (Ziess). Images were analysed using the open source software FIJI[121].

**NOMe-seq library generation and analysis**. Female mESCs were derived by crossing FVB/NJ dams with CAST/EiJ sires. The resultant female mESC lines were expanded and then differentiated using our culture conditions. Cells were transduced with the indicated shRNAs at day 2 of differentiation and samples fixed in 1% formaldehyde at the indicated timepoints. NOMe-seq samples were prepared as described[83], following their protocol for fixed cells. Bisulfite treatment was performed using the EZ DNA Methylation kit (Zymo Research) and sequencing libraries prepared with the Accel-NGS Methyl-Seq DNA Library Kit (Swift Biosciences) and sequenced in-house on the Illumina NextSeq500 platform with 75 bp paired-end reads. Quality control and adapter trimming were performed with fastqc and trim_galore[111], respectively. Using bismark[123], reads were aligned to a version of mm10 with SNPs between FVB/NJ with CAST/EiJ n-masked, created using SNPsplit[114] then bisulfite converted using bismark. Reads were haplotype phased using SNPsplit[114] and methylation calls made with the bismark_methylation_extractor[123]. Methylation calls were filtered for informative CpG and GpC positions using coverage2cytosine with the -nome-seq flag. For analysis of GpC methylation, % methylation was determined at all covered GpC positions and then averaged over 25 positions and normalised using Enrichment normalisation with the Seqmonk package (www.bioinformatics.babraham.ac.uk/projects/seqmonk/). Both heatmap and line plots were produced by averaging over all gene positions in the indicated genomic regions, with line graphs additionally smoothed for clarity using Seqmonk.

**Reporting summary**. Further information on research design is available in the Nature Research Reporting Summary linked to this article.

## Data availability

The data that support this study are available from the corresponding authors upon reasonable request. All next-generation sequencing data generated for this project have been deposited in the Gene Expression Omnibus (GEO) database under accession number GSE137163. Publicly available data were utilised in this study and are available from the GEO database under accession numbers GSE23943 and GSE67299. Source data are provided with this paper.

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

## Acknowledgements

This study was supported by an Australian Research Training Program scholarship (N.J., L.J.G.), a Melbourne Research Scholarship - International (I.W.), the Bellberry-Viertel Senior Medical Research Fellowship (M.E.B.), Sylvia and Charles Viertel Senior Medical Research Fellowship (J.M.P.) and an Australian National Health and Medical Research Council fellowship (M.E.R.). Grant support was provided by the Australian National Health and Medical Research Council (1059624 and 1194345 to M.E.B., 1140976 to M.E.B., M.E.R. and A.K.), the Dyson Bequest and the DHB Foundation. Additional support was provided by the Victorian State Government Operational Infrastructure Support, Australian National Health and Medical Research Council IRIISS grant (9000433). We thank Neil Brockdorff and his laboratory for the kind gift of the inducible *Xist* mESCs.

## Author contributions

A.K., L.J.G., D.J.H. and M.E.B. conceived the study. A.K., N.J., L.J.G., I.W., J.C., C.M.N., J.L., K.A.B., M.I., T.B., A.T.d.F., A.J., T.W. and M.P. performed experiments. A.K., P.F.H., Q.G. and L.W. performed bioinformatic analysis. S.A.K. and P.C.T. provided expertise. A.K., M.E.R. and M.E.B. secured funding. M.E.R., J.M.P. and M.E.B. supervised the project. A.K. and M.E.B. wrote the manuscript with contributions from all authors.

## Competing interests

The authors declare no competing interests.

## Additional information

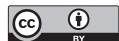

