## [Peer Review File · Nature Communications]

REVIEWER COMMENTS

Reviewer #1 (Remarks to the Author):

In this manuscript, Keniry et al reports a fluorescent reporter system to follow X-inactivation in female mESCs. The authors use this system to perform a mini-screening and identify Smarcc1 and Smarca4 chromatin remodelers as regulators of X-inactivation. The utility of such a report system to follow X-inactivation is enormous. It will control for the appearance of XO mESCs in undifferentiated cultures and would speed up the investigation of X-inactivation in murine cells. Xmas mESCs are indeed a tractable system for future genetic and drug screenings. I don't have concerns about this part of the manuscript. Xmas mESCs was used by the authors to perform a mini-screening, leading to the identification of two members of the chromatin remodeling BAF complex as regulators of X-inactivation. The authors further investigate the mechanisms through which Smarcc1 and Smarca4 affect X-inactivation. The authors claim problems at the establishment phase and suggest a failure to deplete nucleosomes on the inactive X chromosome as the mechanism causing the X-inactivation defective phenotype. However, I have major concerns about this second part of the manuscript. The data presenting seems to suggest a clear problem in Xist coating. The reasons behind this problem were not investigating in detail. The authors rather concentrate their experiments on downstream events that could be mere consequences of that major phenotype.

Major comments:

- The authors claim that Smarcc1 and Smarca4 affect the establishment phase, but not the initiation or maintenance phase. It is unclear what the authors mean by establishment phase (Xist coating / Polycomb recruitment/ Gene silencing). The data in Fig.6a,b points for a very clear phenotype, the absence of a Xist coating in depleted cells for Smarcc1 or Smarca4 (how different are Xist signals at d4-6 in depleted cells from d0 cells in the control situation?). All the phenotypes such as absence of H3K27me3 enrichment, lack of X-linked gene silencing or dynamics of nucleosome depletion at X-linked gene promoters, could all be easily secondary to this phenotype. Failure of Xist coating is the interesting phenotype that should be addressed by the authors. Is that explained by reduction in Xist expression levels/transcription? Is that because the Xist transcript is unstable? Are Smarcc1 and Smarca4 important for Xist spreading along the X chromosome?
- Smarca4 and Smarcc1 proteins are enriched and co-localize with H2Aub at day 5 of differentiation in over 80% of cells (Fig. 6c), while Smarca4 is no longer enriched on H3K27me3-marking Xi at day 6 (extended Fig. 5e). This difference is quite an astonished result. To avoid possible cross-hybridization of antibodies, the authors should repeat the experiments at day 5 using the Smarca4 and Smarcc1 antibodies on its own, or in combination with the H3K27me3.
- A major concern can also be raised about the use of the X-FVB X-CAST system to profile nucleosome depletion. In this system, the X-FVB is slightly skewed towards inactivation of the X-FVB (~75% or so). However, this is far from a perfect 100% skewing, which would be an ideal system for this experiment (XIST-inducible or TSIX-deletion hybrid female cell lines or XIST-inducible XY lines would provide better systems). The authors even comment on reduced coverage of allele-specific data acknowledging the weaknesses of their system. With such weakness it is hard to attribute the accessibility of the promoters of the X-FVB chromosome one day earlier than the X-CAST chromosome to an event anticipating X-linked gene silencing. Moreover, nucleosome opening at promoters seem to be a common feature to autosomes and X chromosomes at day 4-5 of differentiation (Extended Fig.5d). While I think studying nucleosome positioning/depletion in the context of X-inactivation is a very valuable experiment for the X-inactivation field, I think this should be done in a more controlled system.
- Smarcc1 knockdown results in a delay in the opening of promoters in X-FVB chromosome

by one day. Is this a direct or indirect effect? Is Smarcc1 recruited to the Xi at day 4 (by IF)? And more importantly will it be recruited to the gene promoters at this stage (by ChIP-seq)?

Minor comments:

- Introduction Pg3; This sentence might be too simplistic: "Silencing is maintained by DNA methylation and H3K9me3". While this DNA methylation at the promoters of X-linked genes is indeed crucial for maintenance, the role of H3K9me3 seems to be limited to some genes.
- Results Pg4 – For clarification, it should be said in this section that Xmas reporters were done in Bruce4 XY cells and that female Xmas mESCs were then derived from mice. The way it is written gives the erroneous idea that this was done on female mESCs.
- The authors should explain the criteria for choosing 17 candidate genes for the screening.
- The data mentioned in this sentence should be provided as an extended data table: "Despite Smarcc1 and Smarca4 both being members of the same complex, there were no significantly differentially expressed genes in common between Smarcc1 and Smarca4 knockdown groups,...". This data could be interesting for researchers interested in the BAF complex. Moreover, any potential genes affecting XCI or differentiation should be highlighted. This will be a much more rigorous way to display the data than choosing a panel of genes named as XCI genes (Fig.5h) or pluripotency genes (Extended Fig. 4i-j), which is far from being exhaustive.
- Extended Data Fig. 2 – Legend of b) and c) are swapped.
- Extended Data Table 1 – This should be provided on Excel Format to facilitate the analyses by the readers.

Reviewer #2 (Remarks to the Author):

Keniry and colleagues describe the generation of a novel mouse model and ES cell lines that can be implemented for genetic screens and as a replenish tool for X chromosome inactivation studies. XX female cell lines are known to lose one of their 2 X chromosomes during culturing or differentiation. In particular, this model is a useful tool to enrich the ES population for XX cells. However, the mouse model produced and the cell line derived from it has some critical limitations. The use of IRES between the KI gene and the reporter is likely to make the reporter not very bright, as mentioned for the XCI reversal screen. The results from their XCI functional screen are interesting and novel. However, likely, the need for Smarcc1/Smarca4 for cell differentiation, rather than specifically needed for XCI (or Xist upregulation). I believe the author must further be validated or excludes their main hypothesis before this paper can be considered for publication. For example: are Smarcc1/Smarca4 needed for cell differentiation and Xist upregulation? Failure in any of these processes would have an outcome in line with the suggested hypothesis (i.e. Xist needs chromatin remodelling before spreading into genes). The data presented do not fully support the hypothesis made.

Major.

The new reporter line, while useful does not appear to be completely new to me. Similar and more ductile reporters have been already generated. For example, by the Nathans lab (Hu et al, Neuron 2014). Other existing mice model could quickly generate a similar line by crossing (i.e. cell lines used in the XCI reversal screens). Similarly, ES cells and other post-XCI cell lines have been already generated by several labs (including the work cited in this manuscript). Furthermore, the use of an IRES greatly diminishes the reporter messenger level and the power of this mouse model. Having said that, the new mouse model has been well characterized and can be a valuable

tool for the scientific community.

The authors suggest that Smarcc1/Smarca4 are needed to create nucleosome-depleted regions on the future Xi and this is need for Xist silencing activity. However, the data presented seems to support major defects in Xist upregulation (Fig. 6A). I understand why the authors suggest that the failure of creating nucleosome depleted regions is the cause of the lack of Xist spreading, but they do not provide enough evidence to support this. I.e. the very same phenotype can be observed in the case Xist is not properly upregulated, or the cells do not correctly differentiate. In support of their hypothesis, the authors should be able to show Xist localization at nucleosome depleted regions (Xist independent chromatin remodelling during differentiation) using RAP. Alternatively, the most likely hypothesis is that the depletion of Smarcc1/Smarca4 affects Xist upregulation. The authors should try to validate their model using inducible Xist systems in Smarcc1/Smarca4 KD lines, in order to disentangle Xist upregulation from spreading/silencing defects.

The data in fig S1, and the used colour-coding, does not help reach a final conclusion regarding these factors for general cell differentiation. It would be good to show qRT-PCR data for essential differentiation/pluripotency genes such as Rex1, Nanog, Oct4, Sox2, Gata4/6, Nestin etc. I am not familiar with the Euclidean distance method used and this has not been explained well in the text. Ideally, more than 2 replicates would be needed.

In support of the hypothesis that these proteins might be necessary for Xist upregulation (rather than spreading/silencing), neither of these proteins have been found in analogous genetic screens (Moindrot, et al, 2015; Monfort et al., 2015). Please comment in the discussion.

Please also review previous literature regarding the role of Smarcc1/Smarca4 in cell differentiation and add it into the manuscript in the discussion.

-Clarify why these proteins which are part of the same complex and there are no common genes in the deregulated gene lists.

-Clarify why the used RNA FISH cannot discriminate between Xist and Tsix? Probes that can detect Xist or Tsix have been widely used for decades.

Minor

Introduction:

“Silencing is maintained by DNA methylation^{13, 34} and H3K9me^{323, 35}”

-Explain whether the authors are referring to human or mouse XCI as they differ in this regard (i.e. H3K27me3 has a more prominent role in mouse XCI)

“All previous screens for XCI regulators were performed either in differentiated cells for factors that maintain XCI^{12, 23, 35, 82-87}, or non-native systems that instead induce Xist out of context (from an autosome in male cells or prior to exit from pluripotency in female mESCs^{10, 11, 88, 89}). Although small in scale, our screen suggests Xmas mESCs will be suitable for high-throughput screening approaches.”

-Explain/comment on why the Xmas model is better than the ones used in the cited paper. Please update the literature.

Fig. 4 C-F, please explain how statistics were calculated. It is not always clear in the image

where the * stand.

Reviewer #3 (Remarks to the Author):

In the manuscript Keniry et al. established and investigated a novel model system to study X chromosome inactivation. This system has been well and detailed studied and described. In addition, the authors demonstrate that two components of the esBAF complex, namely Smarca4 and Smarcc1 are required for X chromosome inactivation. Eventhough these findings are potentially novel, I have some concerns that might be addressed before acceptance.

Major points:

1. The authors state that they have performed a „genetic screen“ to identify novel factors of XCI. If I understand right, the authors have used shRNAs against 17 genes, which were preselected by previous results. a.) Eventhough 17 genes might be called a genetic screen, the authors should state this limited numbers of genes targeted earlier (e. g. in the abstract and introduction). b.) As a reader it is not entirely clear how and why these 17 genes were selected. The authors might explain there selection criteria better. c.) How can the authors rule out „off target“ effects of shRNAs?

2. The authors state a model „where esBAF is recruited to the future Xi to make it accessible, perhaps to silencing factors...“. To validate this model, I suggest to use ChIP/ChIP Seq of esBAF during different timepoints of XCI plus versus minus Smarcc1. E. g. by Smarca4 antibody?

Minor points:

In Figure 6H the vertical lines of „Xist“ in the lower two subfigures are not at the place where I expected those to be.

Thank you for the opportunity to revise our manuscript titled “Xmas mESC: A female embryonic stem cell system that reveals the BAF complex as a key regulator of the establishment of X chromosome inactivation”, manuscript tracking number NCOMMS-21-01115-T. We were very pleased to hear that Reviewers considered the “utility of such a reporter system to follow X-inactivation is enormous”, that the molecular findings from our “XCI functional screen are interesting and novel” and that our study is “well detailed and described”.

We would like to thank the Reviewers for their careful critique of our work and for their constructive comments. We have found the process to be genuinely helpful and believe it has resulted in a substantial improvement to the manuscript, which includes nine new experiments and one new analysis, resulting in fourteen new figure panels and one new dataset uploaded to GEO. The most significant additions are as follows:

- Immunostaining and survival assays in an inducible *Xist* cell model that confirm failure of XCI, both in the absence of differentiation and with unperturbed *Xist* transcription. This disentangles the role of *Smarca4* and *Smarcc1* in XCI from *Xist* activation and differentiation,
- Volumetric analysis of FISH signals shows early stages of *Xist* spreading are normal.
- Results of the NOME-seq confirmed by an alternative method and in an alternative cell system (MARS-qPCR).
- ChIP-seq for *Smarca4* confirms its presence at X-linked promoters during their relaxation, confirming the direct effect of the BAF complex in the early establishment of XCI.
- A new model figure to better illustrate the timing of key XCI events that fail with depleted *Smarcc1* or *Smarca4*.

Further experimentation and our point-by-point response to the Reviewers comments are detailed below, where their comments are in regular black font and our replies in blue italics. Changes to the text of the manuscript are indicated within the revised manuscript in red.

We hope in combination these changes are considered satisfactory; we believe they have greatly strengthened the paper.

Best wishes,
Andrew Keniry and Marnie Blewitt.

Reviewer #1 (Remarks to the Author):

In this manuscript, Keniry et al reports a fluorescent reporter system to follow X-inactivation in female mESCs. The authors use this system to perform a mini-screening and identify *Smarcc1* and *Smarca4* chromatin remodelers as regulators of X-inactivation. The utility of such a report system to follow X-inactivation is enormous. It will control for the appearance of XO mESCs in undifferentiated cultures and would speed up the investigation of X-inactivation in murine cells. Xmas mESCs are indeed a tractable system for future genetic and drug screenings. I don't have concerns about this part of the manuscript.

Xmas mESCs was used by the authors to perform a mini-screening, leading to the identification of two members of the chromatin remodeling BAF complex as regulators of X-inactivation. The authors further investigate the mechanisms through which *Smarcc1* and *Smarca4* affect X-inactivation. The

authors claim problems at the establishment phase and suggest a failure to deplete nucleosomes on the inactive X chromosome as the mechanism causing the X-inactivation defective phenotype. However, I have major concerns about this second part of the manuscript. The data presenting seems to suggest a clear problem in Xist coating. The reasons behind this problem were not investigating in detail. The authors rather concentrate their experiments on downstream events that could be mere consequences of that major phenotype.

Thank you for this comment and opportunity to improve our analyses. We have included substantial additional data, as detailed below in response to each comment, that better supports our claim that Xist switches on normally, and early coating is normal, but only later after failure of gene silencing do we observe the failure of Xist coating. These new data suggest the Xist coating defect is a consequence of failed gene silencing caused by Smarca4/Smarcc1 depletion, rather than vice versa.

Major comments:

- The authors claim that Smarcc1 and Smarca4 affect the establishment phase, but not the initiation or maintenance phase. It is unclear what the authors mean by establishment phase (Xist coating /Polycomb recruitment/Gene silencing).

We apologise that this was not made clear. We consider the initiation of XCI to be the induction of Xist expression. Beyond this step, we consider Xist coating, polycomb recruitment and gene silencing to be the establishment phase. We now provide an improved model at the end of the paper which defines what we consider establishment of silencing (Fig. 6k). This figure also explains the timing of XCI events that we have measured and how they are perturbed upon depletion of Smarcc1 or Smarca4, which we hope clarifies our data for the reader. Moreover, we have added words of clarification to the introduction (page 3) and discussion (page 13).

The data in Fig. 6a,b points for a very clear phenotype, the absence of a Xist coating in depleted cells for Smarcc1 or Smarca4 (how different are Xist signals at d4-6 in depleted cells from d0 cells in the control situation?). All the phenotypes such as absence of H3K27me3 enrichment, lack of X-linked gene silencing or dynamics of nucleosome depletion at X-linked gene promoters, could all be easily secondary to this phenotype. Failure of Xist coating is the interesting phenotype that should be addressed by the authors.

Thank you for raising this issue. We agree that this area of the manuscript could be improved and have therefore addressed these concerns with additional experiments as well as text and figure changes, as detailed here.

Xist is certainly unable to coat the Xi in Smarcc1 or Smarca4 depleted cells at day 6 of differentiation. Indeed, there are several XCI events that fail in these cells, including H3K27me3 deposition and gene

silencing. However, we consider these failures to be downstream of the failure to open promoters due to the timing at which we observe these events; promoter opening fails at day 4 of differentiation, gene silencing at day 5, Xist coating and H3K27me3 deposition at day 6. We have made these timings clearer by amending the model figure (Fig. 6k), shown above, to highlight the days of differentiation at which key events occur, and added an explanation to the text on page 10.

We now provide Xist RNA FISH at day 0 of differentiation (Extended Data Fig. 5a and below), which shows no background signal at this timepoint, which is an important control as the Reviewer suggests, as it confirms our RNA FISH assays detect induction of Xist transcription. Additionally, we have performed a volumetric analysis of our RNA FISH data (Fig. 6b and below), which shows the Xist signal in control and *Smarcc1* or *Smarca4* depleted cells to be similar at days 4 and 5 of differentiation, suggesting normal Xist spreading at early stages of XCI, at the time when we already observe a failure to open the promoters on the future inactive X, and failed gene silencing. Interestingly, Xist foci are enlarged at day 4 in the *Smarca4* depleted scenario, suggesting accelerated Xist spreading at this timepoint, likely due to the role of *Smarca4* in maintaining pluripotency, which is resolved by day 5. It is only at day 6 when we observe a significant defect in Xist spreading into the characteristic Xist cloud, 1-2 days after we observe failed promoter opening and failed gene silencing. These data suggest *Smarca4* and *Smarcc1* are not required for Xist activation and the early spreading of Xist. These observations are incorporated in the manuscript on page 8.

Is that explained by reduction in Xist expression levels/transcription? Is that because the Xist transcript is unstable?

We have now addressed Xist transcription and Xist stability using an inducible Xist line (Moindrot et al., Cell Reps, 2015), which incorporates a doxycycline inducible Xist transgene on chromosome 17 in male mESCs and allows interrogation of XCI in a system with standardised Xist expression. These cells use the Xist:BglIII:Cherry system which allows for detection of Xist by immunofluorescence. Upon induction of Xist it will form a cloud, which attracts polycomb repressive complexes and gene silencing of chromosome 17. This autosomal gene silencing will then lead to cell death, which can be measured by a loss of Cherry positive cells. We have exploited the Xist cloud formation, H3K27me3 deposition

and cell death (as a measure of gene silencing) in experiments detailed below, allowing us to assess the role of the BAF complex in XCI where *Xist* expression is standardised. Finally, we also use this system to address stability of *Xist*.

First, we tested whether *Xist* was induced normally upon *Smarcc1* or *Smarca4* depletion. qPCR at day 1 post *Xist* induction confirmed *Xist* is correctly induced with *Smarcc1* or *Smarca4* knockdown (Extended Data Fig. 5e and below) in the inducible system.

To test whether features of XCI are perturbed upon *Smarcc1* or *Smarca4* depletion, when *Xist* transcription is normal, we stained for *Xist* and H3K27me3. We found by immunostaining that *Xist* coating and H3K27me3 deposition fail upon *Smarc* knockdown (Extended Data Fig. f,g and below) at day 1 and 2 post induction, as they did in our female mESC models at day 6 of differentiation, despite normal levels of *Xist* transcription in the inducible model. Therefore, these data are consistent with the effect of the *Smarc* proteins being downstream of *Xist* transcription.

It is important to note that the rapidity of induction meant we were unable to analyse the very early stages *Xist* coating, similar to day 4 and 5 of differentiation in the female mESC system. Given at day 1 of *Xist* induction, H3K27me3 is already present at the *Xist* expressing locus, this is likely equivalent to

approximately day 6 in our Xmas system. Therefore, the inducible model is not informative about early spreading of Xist and rather we rely on the differentiating female mESC data presented above.

Finally, these inducible cells die following induction of Xist due to silencing of Chr17 where the transgene resides, at approximately day 2-3 post Xist induction. This gives us an overall measure of the effectiveness of gene silencing. Smarcc1 and Smarca4 depletion confer a substantial survival advantage (Extended Data Fig. 5i and below) when using mCherry positivity by FACS to indicate cells that have induced Xist and go on to survive at day 3 post induction. FACS does not require that the cells form Xist foci, so the data shown in Extended Data Fig. 5f (above) and 5i (below) show different measures.

These data in the inducible Xist cells show that with equivalent Xist transcription levels, we still observe defective inactivation upon Smarc depletion, therefore the role of the Smarc proteins in X inactivation cannot be explained by a difference in Xist transcription. These findings have been incorporated in the manuscript with an extensive rewording on pages 8 and 9.

Given Xist expression was normal in the inducible model, and able to be switched off, we used this model to examine Xist transcript stability. We used qRT-PCR following 1 day of Xist induction and 1 day of doxycycline washout, such that Xist was no longer being transcribed and transcript levels now represent stability. We found Xist levels were reduced in the knockdowns (Extended Data Fig. 5h and below) following dox washout. This result suggests that in the differentiating female mESC cells where we observe a depletion of Xist RNA levels at day 6, Xist is destabilised. Previous reports suggest that Xist can be destabilised when not bound to chromatin (Ng et al, Mol Biol Cell, 2011), which has also been supported in a very recent paper (Rodermund et al., Science, 2021). Therefore, given the failure of Xist to localise to chromatin at this time, we believe the Xist destabilisation is similarly due to failed localisation. These data are described on page 9 of the manuscript.

Finally, we tested whether Smarcc1 or Smarca4 had a role in nucleosome occupancy at the Xist promoter by MARS-qPCR (technique reported in Ohhatta et al., Cell Reps, 2021), which we would

expect if the XCI defect was caused by a change in *Xist* transcription. We found no difference between knockdown and control (Extended Data Fig. 6j and below). This finding has been incorporated in the manuscript on page 11.

Are *Smarcc1* and *Smarca4* important for *Xist* spreading along the X chromosome?

We have performed additional experiments to investigate this possibility. The volumetric analysis of *Xist* RNA FISH foci presented above at day 4, 5 and 6 of differentiation suggests normal spreading of *Xist* in the early stages of XCI where the BAF complex is active (figure shown and discussed above), but a spreading defect at day 6 once the BAF complex is no longer enriched on the inactive X, and after failed gene silencing is apparent. Please see our additional new data via ChIP-seq for *Smarca4* binding the promoters on the inactive X at day 4 of differentiation on the next page of the rebuttal. These data suggest an inactive chromatin state playing a role in *Xist* spreading, a concept that has been published very recently by others (Rodermund et al., Science, 2021).

To further investigate *Xist* spreading, and whether it may occur via open promoters mediated by *Smarca4* and *Smarcc1* we analysed an existing RAP-seq dataset obtained during the timecourse of *Xist*-induced gene silencing in an inducible *Xist* model (Engreitz et al., Science, 2013). We found no indication of *Xist* spreading preferentially through promoters in the early stages of XCI, where our data show open chromatin depends on *Smarca4/Smarcc1* (Figure shown below, but not in the manuscript). We cannot exclude a role for the *Smarcs* in *Xist* spreading however and have therefore made this possibility more explicit in the manuscript through the model that we provide (Fig. 6k and page 2 of rebuttal), in the results section on page 8 and the discussion on page 13.

- *Smarca4* and *Smarcc1* proteins are enriched and co-localize with H2Aub at day 5 of differentiation in over 80% of cells (Fig. 6c), while *Smarca4* is no longer enriched on H3K27me3-marking Xi at day

6 (extended Fig. 5e). This difference is quite an astonished result. To avoid possible cross-hybridization of antibodies, the authors should repeat the experiments at day 5 using the Smarca4 and Smarcc1 antibodies on its own, or in combination with the H3K27me3.

This is an important control that we missed and we thank the reviewer for picking this up, as we have been unable to replicate these data in single stain controls. The original experiment was performed by staining cells with a primary antibody against Smarcc1/a4 followed by the appropriate secondary, then washes, before the H2AK119ub antibody with conjugated fluorophore was added. We are unsure why the background staining for the Smarc proteins was observed, but can only assume there was residual secondary antibody remaining despite washes, that then bound to the H2AK119ub antibody. We are very grateful to have been able to remove this data figure from the manuscript. The removal of these data does not change our interpretation that the Smarc proteins act early on the X chromosome during establishment of XCI, as our H3K27me3 IF shows these proteins to be either present or depleted at the Xi. The difference in our interpretation now is that Smarc proteins are present on the Xi at levels only a little higher than other genomic regions, rather than strongly enriched there as we previously thought. We provide new Smarca4 ChIP-seq which shows the presence of Smarca4 on the X chromosome in females at day 4 of differentiation, above what could be explained by presence on the active X chromosome (Fig 6c, d and below). Note that this experiment was performed in C57BL/6 cells rather than the FVB/CAST model, and so our comparison to male cells, which only have an active X chromosome is necessary to interpret the data, keeping in mind the necessity to expect 2 fold increase in signal on the X in female cells if the binding on the Xi was the same as the Xa. These ChIP data show a mild enrichment of Smarca4 on the Xi which is more in line with the overlap observed by H3K27me3 staining, rather than the large enrichment observed in the now removed H2AK119ub staining. In addition, we were able to see a depletion in Smarca4 binding following Smarcc1 depletion, confirming the effect is mediated by the BAF complex as a whole. We have discussed these results on page 10.

- A major concern can also be raised about the use of the X-FVB X-CAST system to profile nucleosome depletion. In this system, the X-FVB is slightly skewed towards inactivation of the X-FVB (~75% or so). However, this is far from a perfect 100% skewing, which would be an ideal system for this experiment (XIST-inducible or TSIX-deletion hybrid female cell lines or XIST-inducible XY lines would provide better systems). The authors even comment on reduced coverage of allele-specific data acknowledging the weaknesses of their system. With such weakness it is hard to attribute the accessibility of the promoters of the X-FVB chromosome one day earlier than the X-CAST chromosome to an event anticipating X-linked gene silencing. Moreover, nucleosome opening at promoters seem to be a common feature to autosomes and X chromosomes at day 4-5 of differentiation (Extended Fig.5d). While I think studying nucleosome positioning/depletion in the context of X-inactivation is a very valuable experiment for the X-inactivation field, I think this should be done in a more controlled system.

We only provide very limited rationale in the manuscript for why we chose to utilise $X^{FVB}X^{CAST}$ cells with natural skewing of XCI. However, the reason is that we have attempted genomics more than half a dozen times using an *Xist* deletion model to force non-random XCI ($X^{129:\Delta Xist}X^{CAST}$), but found when we freshly isolate such mESC they rapidly become $X^{CAST}O$ upon differentiation and therefore produced artifactual results. We believe this occurs because the 129 allele is naturally favoured as the inactive X and therefore the *Xist* deletion on the 129 allele preferentially pushes these cells towards the $X^{CAST}O$ state, however we do not have conclusive data on this and therefore have not included this detailed explanation in the manuscript. We have now added this concept to the methods section. In comparison, the natural skewing of XCI in $X^{FVB}X^{CAST}$ cells, despite being only 75%, produces very reliable results. The downside of this system is that the magnitude of the effects are minimised due to the only partial skewing, however this is preferential to the artefacts produced by XO cells. It seems that there are a small number of stable cell lines with an *Xist* or *Tsix* knockout, but our preference has been to use freshly isolated ES cells as a renewable resource. We note that the defective XCI we observe in $X^{FVB}X^{CAST}$ cells (RNA-seq and NOME-seq) aligns with data we obtained in C57Bl/6 cells (*Xist* FISH, H3K27me3 IF and Xmas reporter assays), as well as the new data we have provided in the inducible *Xist* male mESC line (*Xist* and H3K27me3 IF and survival assays).

This Reviewer's comments specifically query the NOME-seq data produced in $X^{FVB}X^{CAST}$ cells, therefore we have performed MARS-qPCR (technique reported in Ohhatta et al., Cell Reps, 2021) in C57Bl/6 differentiating mESCs (Extended Data Fig. 6j and again below). This assay also showed promoters to be less open at day 4 in a *Smarcc1* knockout, therefore validating the NOME-seq data using a different cell line without skewed XCI and a different technique. Moreover, we included a *Smarca4* knockdown in this assay, unlike for the NOME-seq, finding the same phenotype as for *Smarcc1*. We have addressed these data on page 11.

- *Smarcc1* knockdown results in a delay in the opening of promoters in X-FVB chromosome by one day. Is this a direct or indirect effect? Is *Smarcc1* recruited to the Xi at day 4 (by IF)? And more importantly will it be recruited to the gene promoters at this stage (by ChIP-seq)?

Immunostaining for inactive X markers such as H3K27me3/H2AK119ub does not produce robust foci at day 4 of differentiation and therefore we cannot easily determine the occupancy of *Smarcc1* or *Smarca4* by this technique. We have instead performed ChIP-seq at day 4 of differentiation for *Smarca4*, for which a more robust antibody exists. We have done this in control and *Smarcc1* knockdown cells, allowing us to determine whether *Smarca4*'s presence on the X chromosome is dependent upon *Smarcc1*. Indeed, we found that *Smarca4* is present at X-linked promoters at day 4 of differentiation and this association is reduced with a *Smarcc1* depletion (Fig. 6e, f and below), suggesting both that *Smarcc1* and *Smarca4* act in complex and that their role at X-linked promoters is direct. Note that this

experiment was performed in C57BL/6 cells rather than the FVB/Cast model, and so our comparison to male cells. This experiment is detailed on page 10 of the manuscript.

Minor comments:

- Introduction Pg3; This sentence might be too simplistic: “Silencing is maintained by DNA methylation and H3K9me3”. While this DNA methylation at the promoters of X-linked genes is indeed crucial for maintenance, the role of H3K9me3 seems to be limited to some genes.

This sentence has been changed and appears on page 3.

- Results Pg4 – For clarification, it should be said in this section that Xmas reporters were done in Bruce4 XY cells and that female Xmas mESCs were then derived from mice. The way it is written gives the erroneous idea that this was done on female mESCs.

We have amended the sentence to include this information, on page 4.

- The authors should explain the criteria for choosing 17 candidate genes for the screening.

The explanation for how the 17 candidate genes were chosen appears on page 7. We did not perform any filtering of the candidates that were reported from the initial screen. We have modified a sentence also on page 7 to make this clear.

- The data mentioned in this sentence should be provided as an extended data table: “Despite Smarcc1 and Smarca4 both being members of the same complex, there were no significantly differentially expressed genes in common between Smarcc1 and Smarca4 knockdown groups,...”. This data could be interesting for researchers interested in the BAF complex. Moreover, any potential genes affecting XCI or differentiation should be highlighted. This will be a much more rigorous way to display the data than choosing a panel of genes named as XCI genes (Fig.5h) or pluripotency genes (Extended Fig. 4i-j), which is far from being exhaustive.

These data are now included as Extended Data Table. 4.

- Extended Data Fig. 2 – Legend of b) and c) are swapped.

This has been amended, thank you.

- Extended Data Table 1 – This should be provided on Excel Format to facilitate the analyses by the readers.

All Extended Data Tables are now in Excel format.

Reviewer #2 (Remarks to the Author):

Keniry and colleagues describe the generation of a novel mouse model and ES cell lines that can be implemented for genetic screens and as a replenish tool for X chromosome inactivation studies. XX female cell lines are known to lose one of their 2 X chromosomes during culturing or differentiation. In particular, this model is a useful tool to enrich the ES population for XX cells. However, the mouse model produced and the cells line derived from it has some critical limitations. The use of IRES between the KI gene and the reporter is likely to make the reporter not very bright, as mentioned for the XCI reversal screen.

The results from their XCI functional screen are interesting and novel. However, likely, the need for Smarcc1/Smarca4 for cell differentiation, rather than specifically needed for XCI (or Xist upregulation). I believe the author must further be validated or excludes their main hypothesis before this paper can be considered for publication. For example: **are Smarcc1/Smarca4 needed for cell differentiation and Xist upregulation?** Failure in any of these processes would have an outcome in line with the suggested hypothesis (i.e. Xist needs chromatin remodelling before spreading into genes). The data presented do not fully support the hypothesis made.

Thank you for highlighting these areas. We have now provided additional data and explanation for cell differentiation and Xist upregulation, as detailed below.

Major.

The new reporter line, while useful does not appear to be completely new to me. Similar and more ductile reporters have been already generated. For example, by the Nathans lab (Hu et al, Neuron 2014). Other existing mice model could quickly generate a similar line by crossing (i.e. cell lines used in the XCI reversal screens). Similarly, ES cells and other post-XCI cell lines have been already generated by several labs (including the work cited in this manuscript). Furthermore, the use of an IRES greatly diminishes the reporter messenger level and the power of this mouse model. Having said that, the new mouse model has been well characterized and can be a valuable tool for the scientific community.

We believe we had cited all similar reporter lines, including the one from the Nathan lab (Wu et al, Neuron 2014) and had attempted to explain our points of difference; specifically other models have either not created mouse strains or do not use endogenous promoters. That said, there has since been another similar reporter line published since (Bauer et al., Nat Comms, 2021) which we have now cited and we have attempted to explain more clearly our points of difference, with the changes appearing on page 4.

It is correct that the use of an IRES diminishes the intensity of our reporter alleles. We were aware of this likelihood when we created the alleles, however believed it was important to ensure that an unmodified Hprt protein was produced, given our intention to create mouse strains. We primarily promote the Xmas system as a tool for FACS analysis and the fluorescence signal is robustly detectable by this method, as demonstrated throughout the manuscript. Detecting the reporter alleles by microscopy, as would be more useful for in vivo studies is more challenging, though possible (see Fig. 2d,e). We have acknowledged this on page 4.

The authors suggest that Smarcc1/Smarca4 are needed to create nucleosome-depleted regions on the future Xi and this is need for Xist silencing activity. However, the data presented seems to support major defects in Xist upregulation (Fig. 6A). I understand why the authors suggest that the failure of creating nucleosome depleted regions is the cause of the lack of Xist spreading, but they do not provide enough evidence to support this. I.e. the very same phenotype can be observed in the case Xist is not properly upregulated, or the cells do not correctly differentiate. In support of their hypothesis, the authors should be able to show Xist localization at nucleosome depleted regions (Xist independent

chromatin remodelling during differentiation) using RAP. Alternatively, the most likely hypothesis is that the depletion of Smarcc1/Smarca4 affects Xist upregulation. The authors should try to validate their model using inducible Xist systems in Smarcc1/Smarca4 KD lines, in order to disentangle Xist upregulation from spreading/silencing defects.

This Reviewer, together with Reviewer 1, has rightly pointed out that this aspect of the paper required improvement and we have therefore provided significant experimentation to sure up our conclusions, as detailed below. To clarify, we do not necessarily believe that nucleosome depleted regions are required for Xist spreading. Our preferred hypothesis is that this allows access to chromatin for silencing complexes. We note both as possibilities as we cannot exclude either at this stage. Xist is certainly unable to coat the Xi in Smarcc1 or Smarca4 depleted cells at day 6 of differentiation, however we consider these failures to be downstream of the failure to open promoters due to the timings at which we observe key XCI events; promoter opening fails at day 4 of differentiation, gene silencing at day 5, Xist coating and H3K27me3 deposition at day 6. We have made the timings of key events clearer by amending the model figure (Fig. 6k and below) to highlight the days of differentiation at which key events occur, and added explanations on page 13.

In support of Xist spreading being downstream of promoter opening, we have performed a volumetric analysis of our RNA FISH data (Fig. 6b and below), which shows the Xist signal in control and Smarcc1 or Smarca4 depleted cells is the same at days 4 and 5 of differentiation, suggesting normal Xist spreading at early stages of XCI. Interestingly, Xist foci upon Smarca4 depletion are enlarged at day 4, suggesting accelerated Xist spreading at this timepoint, likely due to the role of Smarca4 in maintaining pluripotency, which is subsequently resolved by day 5. These observations are incorporated in the manuscript with the following figure and detailed on page 8.

Further support for promoter opening not being involved in Xist spreading comes from analysing Xist localisation to the X chromosome by RAP-seq, as this Reviewer suggested. We analysed an existing RAP-seq dataset obtained during a timecourse of inactivation for an inducible Xist model (Engreitz et al, Science, 2013), and found no indication of Xist spreading preferentially through promoters in the early stages of XCI (Figure shown below, but not in the manuscript). We cannot exclude a role for Smarca4 and Smarcc1 in Xist spreading however, and therefore continue to include this possibility explicitly in the manuscript, but now with some extra explanation in the results on page 8 and the discussion on page 13.

The Reviewer rightly points out that Xist levels are down by qPCR when either Smarcc1 or Smarca4 are depleted, which could cause the defects we observe. We have addressed whether levels of Xist cause the observed failed XCI using an inducible Xist line (Moindrot et al., Cell Reps, 2015), which incorporates a doxycycline inducible Xist transgene on chromosome 17 in male mESCs and allows interrogation of XCI in a system with standardised Xist expression. These cells use the Xist:BglII:Cherry system which allows for detection of Xist by immunofluorescence. Upon induction of Xist it will form a cloud, which attracts polycomb repressive complexes and gene silencing of chromosome 17. This autosomal gene silencing will then lead to cell death, which can be measured by a loss of Cherry positive cells. We have exploited the Xist cloud formation, H3K27me3 deposition and cell death (as a measure of gene silencing) in experiments detailed below, allowing us to assess the role of the BAF complex in XCI where Xist expression is standardised. We note that this is also a differentiation free system for studying XCI, which addresses an additional point raised by the reviewer. Finally, we also use this system to address stability of Xist.

First, we tested whether Xist was induced normally upon Smarcc1 or Smarca4 depletion. qRT-PCR at day 1 post Xist induction confirmed Xist is correctly induced with Smarcc1 or Smarca4 knockdown (Extended Data Fig. 5e and below).

To test whether features of XCI are perturbed upon *Smarcc1* or *Smarca4* depletion, when *Xist* transcription is induced and normal, we stained for *Xist* and H3K27me3. We found by immunostaining that *Xist* coating and H3K27me3 deposition fail upon *Smarcc1* or *Smarca4* knockdown (Extended Data Fig. 5f,g and below) at day 1 and 2 post induction, as they did in our female mESC models, despite normal levels of *Xist* transcription in the inducible model. Therefore, these data suggest the effect of the *Smarc* proteins is independent of *Xist* upregulation.

It is important to note that the rapidity of induction meant we were unable to analyse the very early stages *Xist* coating, similar to day 4 and 5 of differentiation in the *Xmas* cell system. Given at day 1 of *Xist* induction H3K27me3 is already present at the *Xist* expressing locus, this is likely equivalent to approximately day 6 in our *Xmas* system. Therefore, the inducible model is not informative about early spreading of *Xist* and rather we rely on the female differentiating mESC data presented above.

Finally, these inducible cells die following induction of *Xist* due to silencing of *Chr17* where the transgene reside. This gives us an overall measure of the effectiveness of gene silencing. *Smarcc1* or *Smarca4* depletion confer a substantial survival advantage (Extended Data Fig. 6i and below), indicating a failure of silencing.

These data in the inducible *Xist* cells show that with equivalent *Xist* transcription levels, we still observe defective inactivation upon *Smarc* depletion, therefore the role of the *Smarc* proteins in X inactivation cannot be explained by a difference in *Xist* transcription. These findings have been incorporated in the manuscript with an extensive reworking of pages 8 and 9.

Given the decrease in Xist observed in the female mESC system, and the above data suggesting we still have an effect even with normal levels of transcription, we sought to test Xist stability using the inducible Xist model. We used qRT-PCR following 1 day of Xist induction and 1 day of doxycycline washout, such that Xist was no longer being transcribed and transcript levels now represent stability. We found Xist levels were reduced in the knockdowns (Extended Data Fig. 5h and below) following dox washout. This result suggests that in the female mESC where we observe a depletion of Xist RNA levels at day 6, Xist is destabilised. Previous reports suggest that Xist can be destabilised when not bound to chromatin (Ng et al, Mol Biol Cell, 2011), which has also been supported in a very recent paper (Rodermund et al., Science, 2021). Therefore, given the failure of Xist to localise to chromatin at this time, we believe the Xist destabilisation is similarly due to failed localisation. We have incorporated these data on page 9.

The data in fig S1, and the used colour-coding, does not help reach a final conclusion regarding these factors for general cell differentiation. It would be good to show qRT-PCR data for essential differentiation/pluripotency genes such as Rex1, Nanog, Oct4, Sox2, Gata4/6, Nestin etc. I am not familiar with the Euclidean distance method used and this has not been explained well in the text. Ideally, more than 2 replicates would be needed.

We have colour coded the differentiation and pluripotency genes on Extended Data Fig. 4i,j and performed the qRT-PCR suggested, which shows no trend towards slowed differentiation (Extended Data Fig. 4m and below). This data is in support of existing RNA-seq in female cells with Smarcc1.6, Smarcc1.7, Smarca4.4 or Smarca4.6 knockdown and male cells with Smarcc1.6, Smarcc1.7, Smarca4.4 or Smarca4.6 knockdown (Fig. 6 and Extended Data Fig. 4), all of which show no trend towards slowed differentiation; in fact, if anything differentiation appears slightly accelerated (although not statistically significant). We have shown this using Euclidean distance measurements, which is a method to assess correlation between datasets, and has been better explained on page 8 of the manuscript now.

We note that the known role of the BAF complex is to promote pluripotency and self-renewal (Ho et al., PNAS, 2009), such that the effect of depleting these proteins would be to accelerate differentiation and therefore initiation of XCI. Our system was specifically designed to minimise complications with

pluripotency by transducing cells with shRNA during differentiation, after cells exit pluripotency. Indeed, transducing cells even later in differentiation with hairpins against *Smarcc1* or *Smarca4* still leads to failed XCI (Extended Data Fig. 4n). Finally, the new data we provide showing failure to establish key XCI hallmarks upon *Smarcc1* or *Smarca4* knockdown in a differentiation-free inducible *Xist* model completely disentangles the role of the BAF complex in XCI from differentiation. These observations are incorporated in the manuscript on pages 8 and 9.

In support of the hypothesis that these proteins might be necessary for *Xist* upregulation (rather than spreading/silencing), neither of these proteins have been found in analogous genetic screens (Moindrot, et al, 2015; Monfort et al., 2015). Please comment in the discussion.

These screens were performed in pluripotent mESCs with inducible Xist with the Moindrot screen running 7 days and the Monfort screen up to 18 days. This is too long to identify genes that are also required for pluripotency, such as Smarcc1 and Smarca4. Our own experiments in inducible pluripotent cells showed Smarcc1 or Smarca4 depleted cells could be maintained only about 4 days. We have added this observation to the discussion on page 13.

Please also review previous literature regarding the role of *Smarcc1*/*Smarca4* in cell differentiation and add it into the manuscript in the discussion.

We have expanded our discussion of the BAF complex in pluripotency on page 13.

-Clarify why these proteins which are part of the same complex and there are no common genes in the deregulated gene lists.

This is not uncommon with the BAF complex, where depletion of different subunits is known to lead to different chromatin states (Schick et al., Nat Genet, 2021), accessibility and transcription (Schick et al., Nat Genet, 2019). We have referenced this curiosity on page 7 of the manuscript.

-Clarify why the used RNA FISH cannot discriminate between *Xist* and *Tsix*? Probes that can detect *Xist* or *Tsix* have been widely used for decades.

We used cDNA probes to detect Xist because they work robustly in our lab, however they are not strand specific. We provide qRT-PCR to show that Xist is substantially more abundant than Tsix at the critical timepoints, suggesting it is Xist we are detecting. We have now provided Xist RNA FISH at day 0 of differentiation, which does not detect a signal when Xist is not expressed, suggesting that Tsix is not detectable using our cDNA probe and imaging approach (Extended Data Fig. 5a and below). We have incorporated this into the manuscript on page 8.

Minor

Introduction:

“Silencing is maintained by DNA methylation^{13, 34} and H3K9me^{323, 35}”

-Explain whether the authors are referring to human or mouse XCI as they differ in this regard (i.e. H3K27me₃ has a more prominent role in mouse XCI)

We have clarified that we are introducing mouse XCI here, on page 3.

“All previous screens for XCI regulators were performed either in differentiated cells for factors that maintain XCI^{12, 23, 35, 82-87}, or non-native systems that instead induce Xist out of context (from an autosome in male cells or prior to exit from pluripotency in female mESCs^{10, 11, 88, 89}). Although small in scale, our screen suggests Xmas mESCs will be suitable for high-throughput screening approaches.”

-Explain/comment on why the Xmas model is better than the ones used in the cited paper. Please update the literature.

We have now commented on the limitations of prior screens on pages 12 and 13.

Fig. 4 C-F, please explain how statistics were calculated. It is not always clear in the image where the * stand.

We have added an explanation to the Fig. 4 legend.

Reviewer #3 (Remarks to the Author):

In the manuscript Keniry et al. established and investigated a novel model system to study X chromosome inactivation. This system has been well and detailed studied and described. In addition, the authors demonstrate that two components of the esBAF complex, namely Smarca4 and Smarcc1 are required for X chromosome inactivation. Even though these findings are potentially novel, I have some concerns that might be addressed before acceptance.

Major points:

1. The authors state that they have performed a „genetic screen“ to identify novel factors of XCI. If I understand right, the authors have used shRNAs against 17 genes, which were preselected by previous results. a.) Even though 17 genes might be called a genetic screen, the authors should state this limited numbers of genes targeted earlier (e. g. in the abstract and introduction).

We have now made it clear that this was a small-scale screen in both the abstract (page 2) and the introduction (page 3). We developed the Xmas system as a screening platform and have now successfully performed several high throughput screens using these cells. The screen reported here was essentially a pilot screen in preparation for larger screens.

b.) As a reader it is not entirely clear how and why these 17 genes were selected. The authors might explain there selection criteria better.

We have now used more words to explain our selection of the candidate genes, on page 7.

c.) How can the authors rule out „off target“ effects of shRNAs?

To exclude off target effects in all experiments, except the NOME-seq, we have utilised two hairpins against each gene, which beyond the screen and Xmas mESC validation is Smarca4 and Smarcc1. We now provide validation of the NOME-seq by MARS-qPCR, a related protocol to assess nucleosome occupancy(Ohhatta et al., Cell Reps, 2021). See Extended Data Fig. 6j and below. We use two hairpins against each of Smarca4 and Smarcc1 and validate the failure to open promoters at day 4 and then failure to close promoters by day 6 in the Smarcc1 and Smarca4 depleted samples. This is incorporated into the text on page 11.

2. The authors state a model „where esBAF is recruited to the future Xi to make it accessible, perhaps to silencing factors...“. To validate this model, I suggest to use ChIP/ChIP Seq of esBAF during different timepoints of XCI plus versus minus Smarcc1. E. g. by Smarca4 antibody?

This was a valuable suggestion, thank you. We have performed ChIP-seq at day 4 of differentiation for Smarca4, the timepoint at which promoter relaxation occurs. As suggested, this was done in control and Smarcc1 knockdown cells, allowing us to determine whether Smarca4's presence on the X chromosome is dependent on Smarcc1. Indeed, we found that Smarca4 is enriched at X-linked promoters at day 4 of differentiation and this association is reduced with a Smarcc1 depletion (Fig. 6c,d,e,f and below). Note that this experiment was performed in Xmas cells rather than the FVB/Cast model, and so our comparison to male cells, which only have an active X chromosome is necessary to interpret the data.

We observe more peaks on the female X than expected just based on 2 X chromosomes compared with the single male X. This experiment is detailed on page 10.

Minor points:

In Figure 6H the vertical lines of „Xist“ in the lower two subfigures are not at the place where I expected those to be.

This was poor quality control. We now provide an updated model figure (Fig. 6k and below) that we hope better explains the timing at which different key events in XCI occur and fail upon Smarcc1 or Smarca4 depletion.

REVIEWER COMMENTS

Reviewer #1 (Remarks to the Author):

In this revised version, Keniry et al. substantially improved the original manuscript. In general, I am satisfied with the additional data provided and for their answers to my concerns. I would nevertheless highlight a few points for the authors to consider to further improve the manuscript:

1. In Fig.6K, an arrow from day 4 to day 6 would be more adequate to represent the "Establishment" phase, instead of repeating the word "Establishment" three times. Also it would be nice to represent a "Maintenance" stage. The methylation mark on day 6 means H3K27me3, right? This should be mentioned in the legend.
2. In Fig. 6e, the metagene analysis on the X-chromosome would be more informative and controlled if accompanied by a metagene analysis for autosome genes to correct for possible differences in antibody binding/efficiency between different experiments.
3. Accessibility measured by the MARs-qPCR leads to reduced enrichment levels (Fig. S6J), while accessibility by NOME-seq leads to increased levels (Fig. 6H,J). As the read-out of accessibility through these methods lead to values changing in the opposite sense, the authors should explain this in the text, otherwise the reader will get confused and think they are showing opposite results.
4. Although the authors provided more data to show increase nucleosome opening of X-linked promoters early in the establishment phase of X-inactivation, I would be extremely careful as future work will be needed to confirm these results (using systems where Xist expression could be better controlled) and to understand their role in X-inactivation. The effects of these BAF components in X-inactivation could also be multiple and promoter opening might not be the major one. I think this should be discussed more profoundly in the Discussion.

Reviewer #2 (Remarks to the Author):

I am still not entirely convinced about the novelty and the importance of the newly-established mouse/reporter cell line. Many X-linked reporter-gene mice models with endogenous regulation (KI) already exists, in many flavours (phusion/T2A tagging, etc). Just one example, the MeCP2:GFP KI mouse from the Bird lab can easily be crossed with other X-linked KI reporter mice used to study X-linked diseases. These existing models, do not suffer of the reporter intensity diminishment mediated by the IRES. New models have also been published by the Payer's lab, as correctly reported by the authors.

I am still not convinced that the reported phenotype (Xist-mediated gene silencing failure) is due to lack of Xist spreading on the Xi, rather than impairment of Xist upregulation or failure to promoters opening mediated by BAF depletion.

The volumetric analysis shown upon Smarca4 and Smarcc1 (replies to ref1/2), can be misleading as it only considers the cells in which Xist is upregulated, missing important information from the cells that do not upregulate Xist (i.e. see also mCherry IF data).

Xist qPCR data (replies to ref1/2), is not in agreement with the IF data from the Xist:BglII:Cherry, which shows a clear reduction of number of cells initiating XCI. Interestingly, work from the Gerad/Pugh's labs (Dieuleveult 2016) showed Smarca4, as a potential regulator of Xist transcription in mESCs. It is indeed surprising that Xist upregulation is affected by these KDs (IF data) in a Xist inducible cell line (Moindrot et al). However, this can reveal a general role for these chromatin remodelers in transcriptional regulation. For example, Moindrot et al showed the requirement for components of the Mediator complex in Tet-inducible cell lines. Similarly, components of the BAF complex might be essential for both Xist strong upregulation and Tet-O regulation. Furthermore, the survival analysis performed in the Xist:BglII:Cherry, is not suitable to address well the defects of Xist

upregulation vs the defects of gene silencing.

Xist destabilization data looks very convincing (Ext data fig 6), although it does not address the questions above.

MARS qPCR show a clear effect of these KDs on Tsix regulation. Can an increased nucleosome occupancy at Tsix regulatory regions negatively affect endogenous Xist upregulation? Where the Tsix primers were designed?

While I appreciate the figure 6k, I did not suggest that Xist uses open chromatin at promoters. I asked whether Smarca4/Smarcc1 KD affects Xist primary regulation.

Female enrichment of Smarca4 is not very convincing. Showing more female specific peaks can be useful.

Reviewer #3 (Remarks to the Author):

The authors have addressed my major concerns fully. I have no additional comments and recommend the publication of this manuscript.

Thank you for the opportunity to further clarify and improve our manuscript titled “Xmas mESC: A female embryonic stem cell system that reveals the BAF complex as a key regulator of the establishment of X chromosome inactivation”, manuscript tracking number NCOMMS-21-01115-T. The most substantial changes to the manuscript are an experiment indicating normal *Xist* signal when assayed very shortly post induction, as well as an acknowledgement in the discussion that we cannot exclude roles in XCI for the BAF complex beyond promoter opening.

A point-by-point response to the Reviewers comments are detailed below, where their comments are in black font and our replies in blue italics. Changes to the text of the manuscript are indicated within the revised manuscript in red.

We hope that these changes now satisfactorily address all Reviewers concerns.

Best wishes,
Andrew Keniry and Marnie Blewitt.

REVIEWER

COMMENTS

Reviewer #1 (Remarks to the Author):

In this revised version, Keniry et al. substantially improved the original manuscript. In general, I am satisfied with the additional data provided and for their answers to my concerns. I would nevertheless highlight a few points for the authors to consider to further improve the manuscript:

We thank the reviewer for their positive comments and their additional queries which we have actioned and responded to below.

1. In Fig.6K, an arrow from day 4 to day 6 would be more adequate to represent the “Establishment” phase, instead of repeating the word “Establishment” three times. Also it would be nice to represent a “Maintenance” stage. The methylation mark on day 6 means H3K27me3, right? This should be mentioned in the legend.

We have added an arrow to the figure panel, as suggested. Here the methylation mark did indeed refer to H3K27me3, thank you for spotting this omission, we have now amended the figure legend to reflect this. For clarity, we have opted not to represent the maintenance phase of XCI however. This model figure was designed to highlight our findings and therefore only depicts aspects of XCI that we have focused on in this study.

2. In Fig. 6e, the metagene analysis on the X-chromosome would be more informative and controlled if accompanied by a metagene analysis for autosome genes to correct for possible differences in antibody binding/efficiency between different experiments.

Thank you for this suggestion. We have now included the equivalent analysis for autosomal promoters, shown as Extended Data Fig. 6e, and referred to these data in the text on page 10.

3. Accessibility measured by the MARs-qPCR leads to reduced enrichment levels (Fig. S6J), while accessibility by NOME-seq leads to increased levels (Fig. 6H,J). As the read-out of accessibility

through these methods lead to values changing in the opposite sense, the authors should explain this in the text, otherwise the reader will get confused and think they are showing opposite results. *We agree this is quite confusing and that it is best to address this directly to aid the reader. We have now added a sentence to the results on page 11.*

4. Although the authors provided more data to show increase nucleosome opening of X-linked promoters early in the establishment phase of X-inactivation, I would be extremely careful as future work will be needed to confirm these results (using systems where Xist expression could be better controlled) and to understand their role in X-inactivation. The effects of these BAF components in X-inactivation could also be multiple and promoter opening might not be the major one. I think this should be discussed more profoundly in the Discussion.

Thank you for this suggestion, we agree further work will be required to disentangle all possible scenarios for the role of the BAF complex in X inactivation. We have now included an acknowledgement that the BAF complex may have multiple roles in the XCI process, including initiation of Xist (page 14). Please also note new figure panels (Extended Data Fig. 5e,f), which show that inducible Xist upregulation is identical in Smarcc1/a4 depleted cells and control very shortly post Xist induction and prior to spreading, therefore adding weight to our interpretation that Xist is induced correctly but fails to spread.

Reviewer #2 (Remarks to the Author):

I am still not entirely convinced about the novelty and the importance of the newly-established mouse/reporter cell line. Many X-linked reporter-gene mice models with endogenous regulation (KI) already exists, in many flavours (phusion/T2A tagging, etc). Just one example, the MeCP2:GFP KI mouse from the Bird lab can easily be crossed with other X-linked KI reporter mice used to study X-linked diseases. These existing models, do not suffer of the reporter intensity diminishment mediated by the IRES. New models have also been published by the Payer's lab, as correctly reported by the authors.

We accept that novelty in this context is likely subjective and respect the reviewer's opinion. While we agree that published reporters that exist as mice could have been used to produce a female ES cell model for screening, such experiments haven't been reported as such to date. Markers at different loci, rather than both at the same gene, may not provide an equivalent system as each gene is silenced with its own timing. We have ensured we have cited other systems and acknowledged their value, while highlighting our points of difference.

I am still not convinced that the reported phenotype (Xist-mediated gene silencing failure) is due to lack of Xist spreading on the Xi, rather than impairment of Xist upregulation or failure to promoters opening mediated by BAF depletion.

Thank you for this additional opportunity to address this question, we clearly need further clarification in the manuscript. Therefore, we have performed an additional experiment showing that inducible Xist upregulation is identical in Smarcc1/a4 depleted cells and control very shortly post Xist induction and prior to spreading (Extended Data Fig. 5e,f). We have reworded the presentation of results to improve clarity and included a sentence in the discussion which directly acknowledges that we cannot fully exclude a role for the BAF complex in Xist induction. Together we believe these address the reviewer's points and indeed have improved our manuscript. We detail these changes below.

The volumetric analysis shown upon Smarca4 and Smarcc1 (replies to ref1/2), can be misleading as it only considers the cells in which Xist is upregulated, missing important information from the cells that do not upregulate Xist (i.e. see also mCherry IF data).

We agree that in isolation the volumetric analysis is only analysing those cells which switch on Xist, however the scoring data presented in Extended Data Fig. 5b reveals no difference in the proportion of cells that switched on Xist at days 4, 5 or 6; the same stages as we performed the volumetric analysis. Therefore, we believe the volumetric analysis is appropriate and relevant when considering Xist spreading. The scoring analysis presented allows us to query induction of endogenous Xist, which showed no changes, although our analysis is limited to the timepoints we have analysed. We respond below to the Cherry IF data.

Xist qPCR data (replies to ref1/2), is not in agreement with the IF data from the Xist:BglIII:Cherry, which shows a clear reduction of number of cells initiating XCI.

We agree that the Cherry IF data (representing Xist and with fewer Cherry foci when Smarcc1/a4 are depleted) appears to be at odds with the Xist qPCR (no change in Xist levels), both presented in Extended data Figure 5, however they actually measure two different things. We appreciate the opportunity to provide further experimental data and provide greater clarity in how we present these results.

In this inducible system, induction is very much faster than for endogenous X inactivation. By day 1, Xist has already spread to form a cloud characteristic of day 6 of differentiation in the endogenous system. When considering the data already presented, the Cherry IF and Xist qPCR measure two different aspects of Xist. The Cherry IF scores the proportion of cells in which a Cherry focus can be observed, akin to the Xist cloud observed at day 6 of ES cell differentiation. Punctate Cherry foci equivalent to the Xist signal before spreading (day 4 and day 5 in the endogenous system) are not detectable in this assay, likely due to background levels of Cherry. Therefore, the IF was not used to measure Xist induction, but rather Xist spreading. The qPCR for Xist measures Xist levels in all cells, independent of whether Xist spreading is normal or otherwise. Given that there are fewer cells with an Xist cloud, measured by Cherry IF, the equivalent levels of Xist by qPCR suggest Xist is switched on.

A better way to determine correct induction of Xist in the inducible system would have been to perform Xist FISH very quickly post induction of Xist and before spreading occurs. We have now performed this experiment and observe no change in the proportion of nuclei with an Xist signal at 30 minutes post induction, consistent with normal Xist induction in this system. These data are now presented as Extended Data Fig. 5 e, f. Moreover, we have substantially reworded the results to incorporate these new data and more address what each assay measures in the context of these cells with inducible Xist cells (page 9).

In addition to the new results and rewording of the text, we have also incorporated an acknowledgment in the discussion that there may be multiple effects of the BAF complex in the early stages of X inactivation. We hope that the new experiment and improved results and discussion will satisfy the reviewer.

Interestingly, work from the Gerad/Pugh's labs (Dieuleveult 2016) showed Smarca4, as a potential regulator of Xist transcription in mESCs. It is indeed surprising that Xist upregulation is affected by these KDs (IF data) in a Xist inducible cell line (Moindrot et al). However, this can reveal a general role for these chromatin remodelers in transcriptional regulation. For example, Moindrot et al showed the requirement for components of the Mediator complex in Tet-inducible cell lines. Similarly, components of the BAF complex might be essential for both Xist strong upregulation and Tet-O regulation. Furthermore, the survival analysis performed in the Xist:BglII:Cherry, is not suitable to address well the defects of Xist upregulation vs the defects of gene silencing.

The reviewer is correct to point out that our prior experiments did not correctly address concerns that the BAF complex may affect the inducible Xist promoter. We hope that our new Xist FISH data appropriately addresses this concern and given this contend that it is relevant to study the downstream effects of Xist-induced gene silencing in these cells.

Xist destabilization data looks very convincing (Ext data fig 6), although it does not address the questions above.

Thank you, we performed the destabilization tests to specifically address whether Xist was destabilized after induction. We believe it answers this question appropriately and hope that we have now addressed the above questions satisfactorily.

MARS qPCR show a clear effect of these KDs on Tsix regulation. Can an increased nucleosome occupancy at Tsix regulatory regions negatively affect endogenous Xist upregulation? Where the Tsix primers were designed?

Our Tsix primers for the MARS qPCR were taken from Ohatta et al., and are designed around the promoter, ensuring no overlap with Xist. We are not aware of studies that show that the Tsix locus has a role in regulating Xist except via Tsix's transcription. While we observe a more closed chromatin state at the Tsix promoter at day 4 of differentiation, we do not observe any change in the level of Tsix expression (Extended Figure 5d), so this change in nucleosome occupancy does not have a transcriptional consequence on Tsix, and therefore based on our understanding would not impact Xist expression levels, which is expressed from the other allele. Moreover, if increased Tsix nucleosome occupancy was to alter Xist expression, it should result in failure to switch off the second allele of Xist. We did not observe any biallelic Xist expression by RNA FISH at any stage. Finally, it is also relevant to note that Tsix's role is earlier than day 4 of differentiation. We

have added a sentence to the results section to remind readers that Tsix expression is unaltered by Smarcc1/a4 depletion at this stage of differentiation (page 12).

While I appreciate the figure 6k, I did not suggest that Xist uses open chromatin at promoters. I asked whether Smarca4/Smarcc1 KD affects Xist primary regulation.

We hope that the above now addresses the concern regarding primary regulation of Xist.

Female enrichment of Smarca4 is not very convincing. Showing more female specific peaks can be useful.

We have now included additional female specific Smarca4 peak tracks in Extended Data Fig. f,g.

Reviewer #3 (Remarks to the Author):

The authors have addressed my major concerns fully. I have no additional comments and recommend the publication of this manuscript.

Thank you!

REVIEWERS' COMMENTS

Reviewer #1 (Remarks to the Author):

The authors replied satisfactory to my remaining claims and, as such, I will endorse the paper for publication.

Reviewer #2 (Remarks to the Author):

I am satisfied with the current version of the manuscript. The authors' rebuttal letter addresses all my criticisms. Data from the inducible cell line is clear. It is - however -important to highlight the potential role of the BAF complex for Xist regulation in differentiating female ESCs. Thanks for clarifying this in the manuscript.